

Reconciliation of observation- and inventory- based methane emissions for eight large
global emitters
Ana Maria Roxana Petrescu[1], Glen P. Peters[2], Richard Engelen[3], Sander Houweling[1], Dominik Brunner[4], Aki
Tsuruta[5], Bradley Matthews[6], Prabir K. Patra[7,8,9], Dmitry Belikov[9], Rona L. Thompson[10], Lena Höglund-
Isaksson[11], Wenxin Zhang[12], Arjo J. Segers[13], Giuseppe Etiope[14,15], Giancarlo Ciotoli[16,14], Philippe Peylin[17],
Frédéric Chevallier[17], Tuula Aalto[5], Robbie M. Andrew[2], David Bastviken[18], Antoine Berchet[17], Grégoire
Broquet[17], Giulia Conchedda[19], Johannes Gütschow[20], Jean-Matthieu Haussaire[4], Ronny Lauerwald[21], Tiina
Markkanen[5], Jacob C. A. van Peet[1], Isabelle Pison[17], Pierre Regnier[22], Espen Solum[10], Marko Scholze[12], Maria
Tenkanen[5], Francesco N. Tubiello[19], Guido R. van der Werf[23], John R. Worden[24]
[1]Department of Earth Sciences, Vrije Universiteit Amsterdam, 1081HV, Amsterdam, the Netherlands
[2]CICERO Center for International Climate Research, Oslo, Norway
[3]European Centre for Medium-Range Weather Forecasts (ECMWF), Reading, RG2 9AX, UK
[4]Empa, Swiss Federal Laboratories for Materials Science and Technology, 8600 Dübendorf, Switzerland
[5]Finnish Meteorological Institute, P. O. Box 503, FI-00101 Helsinki, Finland
[6]Umweltbundesamt GmbH, Climate change mitigation & emission inventories, 1090, Vienna, Austria
[7]Research Institute for Humanity and Nature, Kyoto 6038047, Japan
[8]Research Institute for Global Change, JAMSTEC, Yokohama 2360001, Japan
[9]Chiba University, 1-33 Yayoicho, Inage Ward, Chiba, 263-8522, Japan
[10]NILU - Norsk Institutt for Luftforskning, Kjeller, Norway
[11]International Institute for Applied Systems Analysis (IIASA), 2361 Laxenburg, Austria
[12]Department of Physical Geography and Ecosystem Science, Lund University, SE-223 62 Lund, Sweden
[13]Department of Climate, Air and Sustainability, TNO, Princetonlaan 6, 3584 CB Utrecht, the Netherlands
[14]Istituto Nazionale di Geofisica e Vulcanologia, Sezione Roma 2, via V. Murata 605, Roma, Italy
[15]Faculty of Environmental Science and Engineering, Babes-Bolyai University, Cluj-Napoca, Romania
[16]Consiglio Nazionale delle Ricerche, Istituto di Geologia Ambientale e Geoingegneria, Via Salaria km 29300,
00015 Monterotondo, Rome, Italy
[17]Laboratoire des Sciences du Climat et de l'Environnement, 91190 Gif-sur-Yvette, France
[18]Department of Thematic Studies – Environmental Change, Linköping University, Sweden
[19]Food and Agriculture Organization of the United Nations, Statistics Division. 00153 Rome, Italy
[20]Climate Resource, Northcote, Australia
[21]Université Paris-Saclay, INRAE, AgroParisTech, UMR ECOSYS, Palaiseau, France
[22]Biogeochemistry and Modeling of the Earth System, Université Libre de Bruxelles, 1050 Bruxelles, Belgium
[23]Meteorology and Air Quality Groep, Wageningen University and Research, Wageningen, the Netherlands





[24]Jet Propulsion Laboratory, California Institute of Technology, Pasadena, CA, USA
*Correspondence to*: A.M. Roxana Petrescu (a.m.r.petrescu@vu.nl)
**Abstract**

Monitoring the spatial distribution and trends in surface greenhouse gas (GHG) fluxes, as well as flux
attribution to natural and anthropogenic processes, is essential to track progress under the Paris Agreement and
to inform its Global Stocktake. This study updates earlier syntheses (Petrescu et al., 2020, 2021, 2023) and
provides a consolidated synthesis of $CH_4$ emissions using bottom-up (BU) and top-down (TD) approaches for
the European Union (EU) and seven additional countries with large anthropogenic and/or natural emissions
(USA, Brazil, China, India, Indonesia, Russia, and the Democratic Republic of Congo (DR Congo)). The work
utilizes updated National GHG Inventories (NGHGIs) reported by Annex I Parties under the United Nations
Framework Convention on Climate Change (UNFCCC) in 2023 and the latest available Biennial Update Reports
(BURs) reported by non-Annex I Parties. The NGHGIs are considered in an integrated analysis that also relies
on independent flux estimates from global inventory datasets, process-based models, inverse modeling and, when
available, respective uncertainties. Whenever possible, it extends the period to 2021. Comparing NGHGIs with
other approaches reveals that differences in the emission sources that are included in the estimate is a key source
of divergence between approaches. A key system boundary difference is whether both anthropogenic and natural
fluxes are included and, if they are, how fluxes belonging to these two sources are grouped/partitioned.
Additionally, the natural fluxes are sensitive to the prior geospatial distribution of emissions in atmospheric
inversions. Over the studied period, the total $CH_4$ emissions in the EU, USA, and Russia show a steady decreasing
trend since 1990, while for the non-EU emitters analyzed in this study, Brazil, China, India, Indonesia, and DR
Congo, $CH_4$ emissions have generally increased.
In the **EU**, the anthropogenic BU approaches are reporting relatively similar mean emissions over 2015
to 2020 of $18.5 \pm 2.7$ Tg $CH_4$ yr$^{-1}$ for EDGAR v7.0, 16 Tg $CH_4$ yr$^{-1}$ for GAINS and 19 Tg $CH_4$ yr$^{-1}$ for FAOSTAT,
with the NGHGI estimates of $15 \pm 1.8$ Tg $CH_4$ yr$^{-1}$. Inversions give higher emission estimates as they include
natural emissions. Over the same period, the three high-resolution regional inversions report a mean emission of
21 (19-25) Tg $CH_4$ yr$^{-1}$, while the mean of six coarser-resolution global inversions results in emission estimates
of 24 (23-25) Tg $CH_4$ yr$^{-1}$. The magnitude of BU natural emissions (peatland and mineral soils, lakes and
reservoirs, geological and biomass burning) accounts for 6.6 Tg $CH_4$ yr$^{-1}$ (Petrescu et al., 2023a) and explains
the differences between the TD inversions and the BU estimates of anthropogenic emissions (including
NGHGIs). For the other Annex I Parties in this study (**USA and Russia**), over 2015 to 2020, the mean of the
four anthropogenic BU approaches reports 18.5 (13-27.9) Tg $CH_4$ yr$^{-1}$ for Russia and 29.1 (23.5- Tg $CH_4$ yr$^{-1}$ for
the USA, against total TD mean estimates of 37 (30-43) Tg $CH_4$ yr$^{-1}$ and 43.4 (42-48) Tg $CH_4$ yr$^{-1}$, respectively.
The averaged BU and TD natural emissions account for 16.2 Tg $CH_4$ yr$^{-1}$ for Russia and 14.6 Tg $CH_4$ yr$^{-1}$ for the
USA, partly explaining the gap between the BU anthropogenic and total TD emissions.
For the **non-Annex I Parties**, anthropogenic $CH_4$ estimates from UNFCCC BURs show large
differences with the other global inventory-based estimates and even more with atmospheric-based ones. This



poses an important potential challenge to monitoring the progress of the global $CH_4$ pledge and the Global
Stocktake, not only from the availability of data but also its accuracy.
By systematically comparing the BU with TD methods, this study provides recommendations for more
robust comparisons of available data sources and hopes to steadily engage more Parties in using observational
methods to complement their UNFCCC inventories, as well as considering their natural emissions. With
anticipated improvements in atmospheric modeling and observations, as well as modeling of natural fluxes,
future development needs to resolve knowledge gaps in both BU and TD approaches and to better quantify
remaining uncertainty. Consequently, TD methods may emerge as a powerful tool for verifying emission
inventories for $CH_4$, and other GHGs and informing international climate policy. The referenced datasets related
to figures are available at https://doi.org/10.5281/zenodo.10276087 (Petrescu et al., 2023b).

## 1. Introduction



In 2021, the NOAA Global Monitoring Laboratory (GML) reported the largest annual increase in
atmospheric $CH_4$ mixing ratios since records began in 1983, with a 17 parts per billion (ppb) value (NOAA
(https://gml.noaa.gov/ccgg/trends_ch4/). In 2022, atmospheric $CH_4$ concentrations averaged 1912 ppb, 162 %
higher than pre-industrial levels. A similar, abnormally large growth rate of 14.8 ppb $yr^{-1}$ was detected from
total column mixing ratio measurements ($XCH_4$) by the Greenhouse Gases Observing Satellite (GOSAT) (Peng
et al., 2022). The drivers of the recent growth are most likely driven primarily by biogenic emissions (Basu et
al., 2022; Lan, et al., 2021a; Lanet al., 2021b; Lan et al., 2022; Nisbet et al., 2016, 2019), with smaller
contributions from increased fossil fuel emissions and a reduced atmospheric sink (Nisbet et al., 2023). These
processes drove the near record increase in atmospheric $CH_4$ growth in 2020, and furthermore outweighed the
slight observed decrease in anthropogenic $CH_4$ emissions accumulated from March–June 2020 as impact of the
COVID-19 slowdown (e.g. China) which might be small relative to the long-term positive trend in emissions.
(McNorton et al., 2022, Peng et al., 2022, Qu et al. 2022).
$CH_4$ in the atmosphere has many different sources, of both natural and anthropogenic origin. The natural
sources of $CH_4$ are dominated by wetlands, while anthropogenic emissions principally come from agricultural
activities (livestock and rice farming), waste management (landfills and water treatment plants) and the
production, transportation, and use of fossil fuels. Most of the agricultural sources are distributed sources, while
the energy-related industrial sources of $CH_4$ are a mix of large point sources, of which some are detectable by
satellite. Smaller point and distributed sources of fugitive emissions (e.g., leaks in pipelines and compression
stations) are more challenging to identify (Rutherford et al., 2021; Omara et al., 2022). While anthropogenic $CH_4$
emissions from fossil fuels, agriculture, and waste can be reduced by mitigation actions, increased natural
emissions lead to different challenges. It has been suggested that fluctuations in natural sources - dominated by
wetlands and open water bodies - were the main reasons for some of the atmospheric $CH_4$ anomalies observed
during the last decades (Rocher-Ros et al., 2023; Zhang et al., 2023; Nisbet et al., 2023; Lunt et al., 2019). Nisbet
et al., 2023 review recent studies, including those which quantified the observed methane growth in the last years.
Using a global inverse analysis of GOSAT satellite observations, it has been shown that increases between 2019-
2020 were in the range of 22-32 Tg $CH_4$ $yr^{-1}$ and were attributed to biogenic sources, half of which took place in
East Africa, as well as Canada (Qu et al., 2022 and Basu et al., 2022).



The contribution of CH₄ to global warming has been estimated to be about 0.5ºC relative to the period
1850–1900 (IPCC, 2021) (Stavert et al., 2022). Methane has a relatively short perturbation lifetime (averaging
12.4 years, Balcombe et al., 2018) and a high global warming potential (86 and 34 for 20- and 100-years times
horizons respectively, compared to that of $CO_2$ emissions, IPCC, 2021, Table 7.15). Given the short lifetime, a
decline in CH₄ emissions will rapidly reduce the global warming contribution from CH₄ and help mitigate the
impact of climate change at decadal time scale (Cain et al., 2021). However, efforts to reduce CH₄ emissions
require a thorough understanding of the dominant CH₄ sources and sinks and their temporal and regional
distribution and trends (Stavert et al., 2022).
The Paris Agreement, a milestone of the UNFCCC to combat climate change and adapt to its effects,
entered into force on November 4, 2016. It asks each signatory to define and communicate its planned climate
actions, known as Nationally Determined Contributions (NDCs), and to report their progress towards their
targets. Next to commitments adopted by countries at COP26, the Global Methane Pledge (GMP) was launched.
The goal of the GMP is to cut anthropogenic CH₄ emissions by at least 30 % by 2030 with respect to 2020 levels,
and is seen as the fastest way to reduce near-term warming and is necessary to keep a 1.5°C temperature limit
within reach. Achieving this goal will drive significant energy security, food security, health, and development
gains. About 150 countries joined this pledge and about fifty already developed national CH₄ action plans or are
in the process of doing so. As agriculture and waste are the main anthropogenic sources for CH₄ emissions, a
GMP Food and Agriculture and Waste pathway was launched at COP27, foreseeing actions that increase
agricultural productivity, reduce emissions from dairy, food loss and waste by supporting small farmers and
increase innovation (https://www.state.gov/global-methane-pledge-from-moment-to-momentum/).
This paper updates previous studies (e.g., Petrescu et al., 2020, 2021, 2023) and aims to inform and attract
attention to the use of the results for diverse climate stakeholder needs beyond research. It deepens the analysis
on sources in the EU and in seven countries that have large anthropogenic and/or natural CH₄ emissions (USA,
Brazil, China, India, Indonesia, Russia and the Democratic Rep. of Congo). It examines both Annex I (EU, USA
and Russia) and non-Annex I estimates from observation-based BU process-based models and inversions-based
TD approaches (using satellite observations) by identifying and explaining differences with official inventory
reports submitted by parties to the UNFCCC. The seven countries were chosen based on location and the
importance / magnitude of their anthropogenic and natural emissions. By using multiple methodologies,
uncertainties can be estimated by looking at the range in both emissions and trends. Starting in 2024, the non-
Annex I Parties must - given they have sufficient capacities - report formal inventories under the Paris
Agreement's Enhanced Transparency Framework following the same guidelines and rules as the Annex I
countries. Furthermore, they will undergo more stringent reviews than those that previously looked at the BURs
and NDCs. This will also allow strengthening the robustness of such comparison exercises when using
independent atmospheric observations in estimating trends and patterns for regional and national CH₄ emissions
(IPCC, 2006).
**2. Methods and data**
**2.1. Verification practices in official UNFCCC NGHGIs**



Quality assurance/quality control (QA/QC) is a key component of NGHGIs development. Verification
is an additional step and refers specifically to methods that are external to the inventory and apply independent
data. There are two main methods of verification: 1) independent inventory-based estimates, 2) observation-
based emission estimates.

A challenge with comparisons against *independent inventory-based estimates* is that none are truly
independent as they may rely on, for example, the same activity data reported by a country (Andrew 2020).
Experience has shown that performing detailed comparisons (Petrescu et al., 2021, 2023) can help clarify
differences in system boundaries or even identify errors (Andrew 2020). Improving independent emission
inventories also has value, as these are often used in global studies where common methods across all countries
are desired.

*Observation-based estimates* use observations of atmospheric concentrations and prior fluxes that are
then coupled to a transport model. These methods are more complex, computationally expensive, but make use
of observational information that is independent from emission inventories.

Since most developed countries have reported UNFCCC inventories for decades and these have been
continually reviewed and refined, the focus of this work is on observation-based estimates. As an increasing
number of developing countries begin more detailed and frequent reporting, comparisons with independent TD
approaches will be an important method of verification for those countries.

The 2019 refinement of the 2006 IPCC guidelines highlight notable advances in the application of
inverse models of atmospheric transport for estimating emissions at the national scale. Building on this progress,
they extend the guidance on the use of atmospheric measurements for verification (IPCC, 2019). There are
several countries that currently use atmospheric measurements for verification of parts of their inventories.
Australia (Luhar et al 2020, Australian NIR, 2023) and New Zealand (Geddes et al., 2021) have estimated
regional $CH_4$ emissions to help better understand the methods and their potential. Germany performs various
cross validation checks with available data (German NIR, 2023), some of which are based on observations. The
UK and Switzerland (Annex 6 CHE NIR, 2023) have developed more comprehensive methods based on
inversion modeling, covering several GHGs in addition to $CH_4$. Building on modeling experience, the country
reporting confirms that most potential lies in using observations to verify fluorinated gases (Annex 6 UK NIR,
2023), but the large uncertainty in $CH_4$ emissions gives the potential for verification if a sufficient observation
network is used in inversion modeling (Bergamaschi et al., 2018, Thompson et al., 2014).

While inversions of $CH_4$ fluxes are associated with significant uncertainty, so are NGHGI estimates of
anthropogenic $CH_4$ emissions. Furthermore, inversions can provide information on subannual and subnational
variations in time and space that may indicate differences in source sector emission estimates. In geographic
areas with sufficiently dense ground-based observation networks, the inversions will have more value.

**2.2. Anthropogenic $CH_4$ emissions from the NGHGIs**


Annex I countries report their annual GHG emissions to the UNFCCC in the so-called Common
Reporting Format (CRFs) data tables and National Inventory Reports (NIRs). Here, anthropogenic $CH_4$
emissions from the five UNFCCC sectors (incl. LULUCF) are grouped together. As part of the LULUCF sector,
we also have the $CH_4$ emissions from wetlands, which according to the are defined as managed "where the water





table is artificially changed (i.e. lowered or raised) or those created through human activity (e.g. damming a
river) and that do not fall into Forest Land, Cropland, or Grassland categories (IPCC, 2014). Reporting $CH_4$
emissions from managed wetlands are not mandatory, but if done, parties are encouraged to make use of the
2013 IPCC Wetlands supplement (IPCC, 2014). In the EU, if Member States report these emissions, they report
not only restored (rewetted) wetlands but also emissions from drained organic and mineral soils (e.g. peatlands,
ditches etc.). These are not large by magnitude but are large by area in the Nordic countries. According to NGHGI
data, in 2021, managed wetlands in the EU for which emissions were reported under the LULUCF (CRF Table
4(II) and Summary 1.As2 accessible for each EU[1]) summed up to 0.21 Tg $CH_4$ yr$^{-1}$ Furthermore, the NGHGIs do
not include any lateral fluxes from inland waters but do include biomass burning anthropogenic emissions
reported under the LULUCF sector.
The presented uncertainties in the $CH_4$ emission levels of the individual countries and the EU are not
always reported in a complete and harmonized format, and therefore were calculated applying gap-filling and
harmonization procedures that are used to compile the EU GHG inventory reported under  UNFCCC (EU NIR,
2023) (see SI and Appendix A1.1 in Petrescu et al., 2023a). The EU uncertainty analysis reported in the bloc's
National Inventory Report (NIR) is based on country-level, Approach 1 uncertainty estimates (IPPC, 2006, Vol.
1, Chap. 3) that are reported by EU Member States, previously under Article 7(1)(p) of Regulation (EU) 525/2013
and since 2023 under Article 26(3) and Annex V(Part 1)(m) of the Governance Regulation (EU) 2018/1999.
Non-Annex I countries report their updated NGHGIs to the UNFCCC, including a national inventory
report and information on mitigation actions, needs and support received in the so-called Biennial Update Reports
(BURs). In this study, Brazil, China, Indonesia, India and the Democratic Rep. of Congo (DR Congo) were
investigated. For Brazil, information from its fourth biennial update report (4[th] BUR) (Brazil, 2020) that give
both total and sectoral split emission values for years 1994, 2000, 2010, 2012, 2015 and 2016, were used. For
China, information from its second biennial update report (2[nd] BUR) Tables 2-10, 2-13, 2-14, 2-15, and 2-16
(China, 2019) were used. The information was available for both total and sectoral split emission values for 1994,
2005, 2010 and 2014. Uncertainties for 2014 are available in Table 2-12. Indonesia submitted its third biennial
update report (3[rd] BUR) in 2021 (Indonesia, 2021). Indonesian total $CH_4$ emissions time series per sector as
reported by the 2[nd]  UNFCCC BUR (2001-2016) and revised 3[rd]  BUR (2000 and 2019, Table 2). For 2017 and
2018, agricultural $CH_4$ emissions were detailed by the 3[rd]  BUR. Data uncertainty for 2019 activity and EFs are
the same as reported in the 2[nd] BUR (2018). The result of the uncertainty analysis showed that the overall
uncertainty of Indonesia's National GHG inventory with AFOLU (including peat fires) for 2000 and 2019 were
approximately 20.0% and 19.9%, respectively. A smaller uncertainty, 10.4 % for 2000 and 13.8 % for 2019,
occurred when the FOLU (including forest fires), was excluded from the analysis. This shows that Indonesian
emission inventories are highly uncertain when forest fires are included in the analysis. The DR Congo submitted
its first BUR in 2022. However, we only used total values reported for 2000-2018 (Table 12 Congo, 2022). India
has submitted three BURs and information on sectoral $CH_4$ emissions are in each of them only for one year. We

---

[1] https://unfccc.int/process-and-meetings/transparency-and-reporting/reporting-and-review-under-the-convention/greenhouse-gas-inventories-annex-i-parties/national-inventory-submissions-2019



compiled information for 2010 from the first BUR (India, 2016), for 2014 from the second BUR (India, 2018)
and for 2016 from the third and latest BUR (India, 2021).
**2.3. Other CH$_4$ data sources and estimation approaches**

The CH$_4$ emissions in the EU and non-Annex I countries used in the atmospheric inversions and
anthropogenic and natural emissions estimates from various BU approaches and inventories (i.e., UNFCCC
CRFs and BURs) covering specific products, sectors and activities are summarized in Table 1. The data and the
detailed description of most products (Tables S1 and S2, Supplementary Information) span the period from 1990
to 2021, with some of the data only available for shorter time periods. The estimates are available both from
peer-reviewed literature and from unpublished research results from the VERIFY and CoCO$_2$ projects
(Supplementary Information, SI) and in this work they are compared with NGHGIs reported in 2023 (time series
for all (Annex I) or some years (non-Annex I) of the 1990-2021 period). The BU anthropogenic sources are from
UNFCCC NGHGIs and three global inventory datasets/models: EDGARv7.0, FAOSTAT and GAINS. In this
synthesis, data from FAOSTAT (Tubiello et al., 2022; FAO, 2023) is included for all economic sectors: Energy,
IPPU, Waste and Other, and are sourced from the PRIMAP-hist v2.4 dataset (Gütschow et al., 2022) to build
emissions indicators on agrifood systems and on the entire economy. Emission totals from the agrifood domain
are computed following the Tier 1 methods of the Intergovernmental Panel on Climate Change (IPCC)
Guidelines for NGHGIs. Agrifood systems emissions in FAOSTAT are largely based on FAO crop, livestock
and land use statistics (Tubiello et al., 2022; FAO, 2023). They are complemented with activity data from the
UN Statistical Division (UNSD), the International Energy Agency (IEA) and with geospatial information on
drained organic soils and biomass fires (Conchedda and Tubiello, 2020; Prosperi et al., 2020).
The analysis focuses on both total and sectoral or partitioned information from both BU and TD
estimates. As detailed in Table 1, not all inversions distinguish between sources, however in the following
sections we discuss comparability between BU and TD for both total and partitioned results.
*Table 1: Sectors included in this study and data sources providing estimates for these sectors. CAMS stands for*
*Copernicus Atmosphere Monitoring Service. References to data products are found in Table 2 Petrescu et al.,*
*2023a and Table S1 and S2, SI.*

| Anthropogenic (BU)[2] CH$_4$ | Natural (BU)[3] CH$_4$ | Regional TD CH$_4$ | Global TD CH$_4$ |
|---|---|---|---|
| 1. Energy: UNFCCC NGHGI (CRFs and BURs), GAINS, EDGAR v7.0, FAOSTAT-PRIMAP | Wetlands<br><br>**EU**: JSBACH-HIMMELI<br><br>**Global**: LPJ-GUESS | **No partitions – total emissions**<br><br>FLEXkF_v2023 | **Totals and**<br><br>**partitioned emissions:** |
| 2. Industrial Products and Products in Use (IPPU): UNFCCC NGHGI (CRFs | | | |

---

[2] For consistency with the NGHGI, here we refer to the five reporting sectors as defined by the UNFCCC and the Paris Agreement decision (18/CMP.1),the IPCC Guidelines (IPCC, 2006), and their Refinement (IPCC, 2019a), with the only exception that the latest IPCC Refinement groups together Agriculture and LULUCF sectors in one sector (Agriculture, Forestry and Other land Use - AFOLU).

[3] The term **natural** refers here to unmanaged natural CH$_4$ emissions (peatlands, mineral soils, geological, inland waters and biomass burning) not reported under the anthropogenic UNFCCC LULUCF sector.





| | | | |
|---|---|---|---|
| and BURs), EDGAR v7.0, FAOSTAT-PRIMAP | Peatlands, mineral soils: **EU**: JSBACH-HIMMELI **Global**: LPJ-GUESS | CIF-FLEXPARTv10.4 CIF-CHIMERE | MIROC4-ACTM (control and OH varying runs) |
| 3. Agriculture: UNFCCC NGHGI (CRFs and BURs), GAINS, EDGAR v7.0, FAOSTAT | | | CAMSv21r1 (NOAA and NOAA_GOSAT runs) |
| 4. LULUCF: UNFCCC NGHGI (CRFs and BURs) and FAOSTAT | Inland waters fluxes **EU**: lakes, rivers and reservoirs (RECCAP2) **Global:** lakes and reservoirs ORNL DAAC | | TM5-4DVAR (TROPOMI) |
| 5. Waste: UNFCCC NGHGI (CRFs and BURs), GAINS, EDGAR v7.0, FAOSTAT-PRIMAP | Geological fluxes updated activity (see SI) | | CTE-CH$_4$ (GCP2021) CEOS (GOSAT) |
| | Biomass burning (GFEDv4.1s) | | GEOS-Chem CTM (TROPOMI) for USA only) |

note: Not all models have a version id. Those that have, are used in previous syntheses (Petrescu et al., 2021 and 2023a).
We define as natural sources, all sources which do not belong to the anthropogenic partition (Table 2).
The BU natural components for the EU were computed as the sum of the VERIFY products (biomass burning,
inland waters and undisturbed peatlands plus mineral soils as described in Petrescu et al., 2021 and 2023) and
geological emissions (Etiope et al. 2019) updated for the VERIFY project. For the seven non-EU emitters, the
BU natural fluxes are the sum of wetland emissions (LPJ-GUESS), lake and reservoir emissions (ORNL DAAC),
biomass burning emissions (GFED4.1s) and geological emissions (updated activity, SI). The TD natural global
estimates were calculated as the sum of all natural partitions reported by the inversions. Adjustments were made
to have a consistent comparison between partitions, adding the missing ones from the BU estimates (Table 4).
The error bar on the TD natural represents the range of the min/max between inversion estimates.
The total regional TD estimates (for EU) and their uncertainties were calculated as the mean and
min/max range between FLEXkF_v2023, CIF-FLEXPART and CIF-CHIMERE inversions (see Priors table in
Petrescu et al., 2023b). For the USA, we considered as a regional estimate the optimized emissions from the
GEOS-Chem CTM (based on TROPOMI data for 2019) from Nesser et al., 2023, with the range from the eight
members of the inversion ensemble shown as uncertainty (Table 2 in Nesser et al., 2023).
For all countries, the total global TD inversion estimates (time series) and uncertainties were calculated
over the period 2015-2021 using the mean and min/max between CTE-GCP2021, MIROC4-ACTM both runs,
CAMS v21r1 (both runs), and TM5-4DVAR (TROPOMI based). CEOS (GOSAT) provided an estimate only
for 2019.
The units used in this paper are metric tons (t) [1kt = 10$^9$ g; 1Mt (Tg) = 10$^{12}$ g] of CH$_4$. The referenced
data for replicability purposes are available for download at https://doi.org/10.5281/zenodo.10276087 (Petrescu
et al., 2023b). Upon request, the computer code for plotting figures in the same style and layout can be provided.
Throughout the paper and mostly for the complex figures, the following ISO3 country codes are used: BRA



(Brazil), CHN (China), IDN (Indonesia), IND (India), RUS (Russia) and COD (DR Congo). As before in the
text, the European Union consists of 27 MS, excludes the UK and is further abbreviated as EU.
**3. Results**

**3.1. NGHGI official reported estimates (UNFCCC)**

Figure 1 presents anthropogenic $CH_4$ emissions reported to the UNFCCC in 2023 from the NGHGI
CRFs (EU, USA and Russia) and BURs (Brazil (4th in 2021), China (2nd in 2019), Indonesia (3rd in 2021), DR
Congo (1st in 2022) and India (all three BURs). The following section provides additional details for all the
countries.
For the EU, the total $CH_4$ emissions in 2021 account for $14.8 \pm 1.8$ Tg $CH_4$ yr$^{-1}$ and represent 12.8 % of
the total EU greenhouse gas emissions (in $CO_2$e, GWP 100 years, IPCC AR5[4]). $CH_4$ emissions are predominantly
from agriculture (Figure 1, brown), a sector which in 2021 accounted for 8.3 Tg $CH_4$ yr$^{-1}$ $\pm$ 0.8 Tg $CH_4$ yr$^{-1}$ or 56
% of the total EU $CH_4$ emissions (incl. LULUCF). Anthropogenic NGHGI $CH_4$ emissions from the LULUCF
sector are very small for the EU e.g., 0.5 Tg $CH_4$ yr$^{-1}$ or 3 % in 2021, including emissions from biomass burning.
The EU data from Figure 1 shows steadily decreasing trends for all sectors with respect to the 1990 $CH_4$ levels.
The reduction in total $CH_4$ emissions in 2021 with respect to 1990 is 8.9 Tg $CH_4$ yr$^{-1}$ (37 %) at an average yearly
rate of -1%.
Between 1990-2021, the reported USA $CH_4$ emissions show a small decrease of 4.6 Tg $CH_4$ yr$^{-1}$, more
pronounced for the last two years (2020-2021), with an average reduction of -0.5 % per year (Fig. 1 black dotted
line). In the USA, the largest share of emissions comes from the Energy sector, for which, next to IPPU and
Waste, the highest reduction shares were registered (42%, 34 % and 26 %, respectively) while emissions from
Agriculture and LULUCF increases of 16 % and 23 %, respectively, were registered. After a notable decrease of
1.5 Tg registered in 2016 compared to 2015, emissions picked up again and had a second decreasing trend in
2020 and 2021, possibly due to the COVID pandemic. Overall, reported data indicates that reductions in the
USA $CH_4$ emissions have been slower than that in the EU.

---

[4] IPCC AR4 GWP 100 values are still used by the Member States in their NGHGI reporting to the UNFCCC.

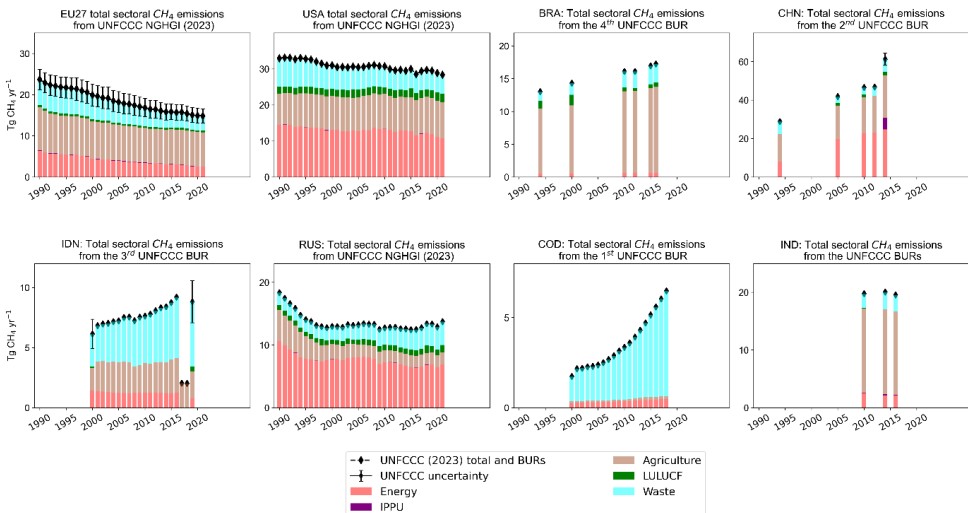


*Figure 1: Total and sectoral CH$_4$ emissions (incl. LULUCF) from the UNFCCC NGHGI (2023) CRFs (EU, USA*

*and Russia) and BURs (Brazil (4$^{th}$ in 2021), China (2$^{nd}$ in 2019), Indonesia (3$^{rd}$ in 2021), DR Congo (1$^{st}$ in 2022)*

*and India (all three BURs:2016, 2018 and 2021). The relative error on the UNFCCC value represents the*

*NGHGI (2023) reported uncertainties computed with the error propagation method (95% confidence interval)*

*and gap-filled to provide respective estimates for each year. Information on Indonesian sectoral CH$_4$ emissions*

*in 2017 and 2018 are only available for Agriculture. In 2014, China reported uncertainty as well (min 5.2 %*

*and max 5.3 %). The overall uncertainty of Indonesia's National GHG inventory with AFOLU (including peat*

*fires) for 2000 and 2019 were approximately 20.0% and 19.9%, respectively.*

The trend in total CH$_4$ emissions in *Brazil* is increasing strongly, with 32.5 % more emissions in 2016
compared to 1994, registering a maximum annual growth rate of +22 % in 2010 compared to 2000, and a
minimum annual increase rate of +1% in 2016 compared to 2015. The agricultural sector (76 % of the total) was
the main driver of the growth, followed by the waste sector (16 % of the total). There are only small CH$_4$
emissions from the Energy sector (some oil and gas activities). The Brazilian agricultural CH$_4$ emissions are the
highest compared to all other countries on a per capita basis (see Figure 2).



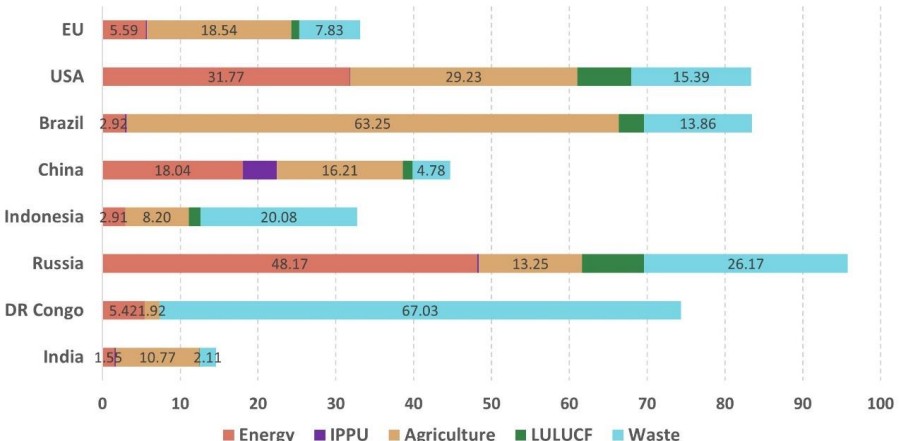

*Figure 2: Per capita emissions (kg) intensity per sector based (IPCC, 2006) on reported emissions and population data from the last reported NIRs as follows: 2021 for the EU, USA and Russia, 2016 for Brazil and India, 2014 for China, 2019 for Indonesia and 2018 for DR Congo.*

*China's* total $CH_4$ emissions are much larger than the emissions reported by many developed countries or the entire EU (see Figure 1), but on a per capita basis it is only in fifth place (Figure 2), with the highest contribution from the Energy sector (third place after Russia and the USA). The rapid growth of China's coal demand has important implications for $CH_4$ emissions from coal mining or coal mine methane (CMM) emissions (Gao et al., 2020). The Energy sector is the largest component of Chinese emissions (40 % in 2015), followed by Agriculture (36 %). The second Chinese UNFCCC BUR $CH_4$ data shows increasing trends in total $CH_4$ emissions with an increase reported in 2014 of 113 % compared to 1994, corresponding to 32 Tg $CH_4$. The Energy and Agriculture sectors have increased by 214 % and 54 % in 2014 compared to 1994.

*Indonesia's* 3rd BUR data (2000 and 2019) show increasing trends in total $CH_4$ emissions. The time series 2001-2006 belongs to the 2nd BUR submitted in 2018. In 2019, Indonesian $CH_4$ emissions have increased by +44 % compared to 2000, corresponding to 2.6 Tg $CH_4$ yr$^{-1}$, an average yearly increase of 3 %, and the sector which contributes the most to this increase is the Waste sector, which nearly doubled its emissions in 2019 compared to 2000, which is also seen in the per capita contribution (Figure 2). According to Qonitan et al., 2021, the major solid waste source in Indonesia is the household sector, which contributed 44-75% to total waste generated. The composition of municipal waste consists of 43.78% of food waste, 16.05% of paper, and 14.08% of plastics. $CH_4$ emissions from the other sectors remained nearly constant.

*Russian* $CH_4$ emissions have decreased by -25 % from 1990 to 2021, but most of this decrease happened during the collapse of the former Soviet Union. Since 2000, $CH_4$ emissions have remained rather low. The decline seen between 1990-2000 is primarily due to the Agricultural sector (-52 %) and Energy (-27 %). At the same time, the Waste sector started to increase its emissions (6 %). Between 2001-2021, the $CH_4$ emissions from the Agriculture and Energy sectors continue to decrease (by 17 % and 11 %, respectively), while the emissions from



the Waste sector register an additional 76 % increase. IPPU emissions increased by 85 % but remain negligible
compared to other sectors. Since the 2000s, also LULUCF emissions have increased by 53 %.

For its first BUR, *DR Congo* submitted emissions from Energy, AFOLU (Agriculture plus LULUCF)

and Waste for 2000-2018. Since 2000, the DR Congo total $CH_4$ emissions have increased by a factor of four.
Most of the $CH_4$ emissions are reported for the Waste sector, and account for 90 % of the total emissions. The
high percentage of waste emissions in DR Congo is seen as well in the per capita Figure 2. Assè-Wassa Sama
and Berenger, 2023 report confirm that between 2000 and 2021, $CH_4$ emissions, which in 2021 represent in DR
Congo ~97% of total waste generated emissions, grew at a rate of 4 % $yr^{-1}$, compared with 2.7 % $yr^{-1}$ for total
emissions. This increase was driven by the increase in emissions caused by solid waste disposal (+6.2 %). The
$CH_4$ waste emissions come mainly from the treatment and discharge of wastewater (69 % in 2021, compared
with 80 % in 2000), followed by the elimination of solid waste (31 % in 2021, compared with 20 % in 2000).
The weight of emissions caused by the elimination of solid waste in the sector's total emissions has nevertheless
increased by 11 percentage points between 2000 and 2021 (Assè-Wassa Sama and Berenger, 2023).

Each of *India's* BURs provide detailed information on sectoral $CH_4$ emissions only for one year. Most

of the emissions in India belong to the Agriculture sector, amounting to almost 15 Tg $CH_4$ $yr^{-1}$ (in 2016),
representing 74 % of the total anthropogenic emissions. From only three years of reported data, there is no clear
notable trend.

## 360    3.2. NGHGI compared to other bottom-up estimates

Figure 3 shows UNFCCC (CRFs and BURs) estimates from EU and seven non-EU countries

compared to global bottom-up inventories.

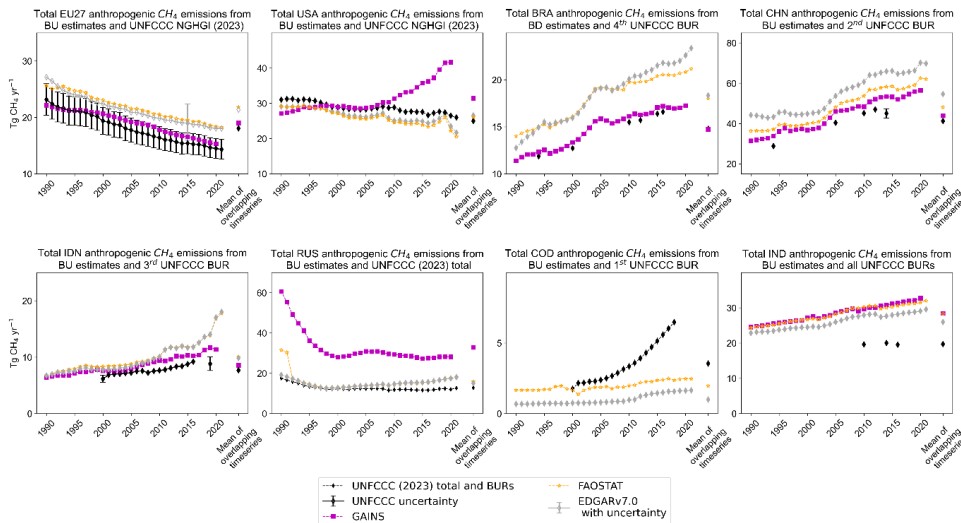


*Figure 3: Total anthropogenic $CH_4$ emissions (excl. LULUCF) from bottom-up (BU) inventories as: UNFCCC*

*NGHGIs (2023) of CRFs (EU, USA and Russia) and BURs (Brazil (4th in 2021), China (2nd in 2019), Indonesia*

*(3rd in 2021), DR Congo (1st in 2022), India (all three BURs:2016, 2018 and 2021) and three other global*





*datasets: EDGARv7.0, GAINS (no IPPU) and FAOSTAT (PRIMAP based, except for AFOLU). The relative error*
*on the UNFCCC value represents the NGHGI (2023) reported uncertainties computed with the error*
*propagation method (95% confidence interval) and gap-filled to provide respective estimates for each year.*
*China and Indonesia report uncertainties, for 2014 and 2000 and 2019 respectively (BUR). Total COD*
*UNFCCC BUR emissions do not include IPPU. The EDGARVv7.0 uncertainty is only for 2015 and was*
*calculated according to Solazzo et al., 2021 for EDGARv5.0. The mean of overlapping time series was calculated*
*for 1990-last available year as following: 2021 for UNFCCC NGHGI (2023), EDGARv7.0 and FAOSTAT and*
*2020 for GAINS.*

From Figure 3, it is notable that, except for the EU and USA which show decreasing trends in emissions
from all data sets (USA except for GAINS), all the other countries show increasing trends. The match between
UNFCCC reported emissions and all the data sources is satisfactory, with few exceptions.
For the EU, the difference between the UNFCCC NGHGI 1990-2020 average and the other three data
sets is less than 5 %. As previously discussed, the inventory-based data sources are consistent with each other
for capturing recent $CH_4$ emission reductions, but they are not independent because they use similar methodology
with different versions of the same AD (Petrescu et al., 2020, Figure 4).
For the USA, GAINS reports high emissions after 2010, with strong growth. This divergence is largely
found in the Energy sector, resulting from the EFs used for conventional gas production as well as for
unconventional shale gas extraction, which has increased rapidly since 2006 due to the development of hydraulic
fracturing technology (Supplementary Figure S6-1 in Höglund-Isaksson et al., 2020). The high share of
emissions from unconventional shale gas can be explained by the GAINS EFs which, in the absence of published
factors, are derived from the residual emissions after having subtracted estimated emissions for oil production
and conventional gas production from the total upstream emission estimated by Alvarez et al., (2018, Table 1)
As Alvarez et al. 2018 do not specify emission factors by type of gas produced, GAINSv4 splits it based on
activity data from other references (IEA-WEO, 2018 and EIA, 2019). On the other hand, the NGHGI EF seems
to be too low, and this is reflected by the low oil and gas emissions reported by the USEPA, 2017 in 2015,
compared to Alvarez et al., 2018 (Supplementary Table S6-3, Höglund-Isaksson et al., 2020). For the USA, total
gas production increased by 47 % between 2006 and 2017. Revisions for the agricultural livestock emissions
concern updates of AD and reported EFs to statistics from FAOSTAT (2018) and CRFs UNFCCC (2016; 2018),
and a review of available technical abatement options for $CH_4$.
For Brazil, UNFCCC and GAINS report emissions of similar magnitudes and trends. The EDGARv7.0
and FAOSTAT report on average around 23 % more emissions for the 1990-2021 period, but closely follow the
NGHGIs trends. The similarity between trends could be explained by the use of the same EFs following Tier-1
IPCC 2006 Guidelines and UNFCCC NIRs (Janssens-Maenhout et al., 2019), while the higher emissions could
appear when using different AD information.
For China the inventory estimates agree with the BUR reported data, with EDGARv7.0 showing the
highest estimates. According to both GAINS and EDGARv7.0, the primary drivers for growth in Chinese $CH_4$





emissions are due to a mix of sources, mainly from the IPCC 2006 sector 1.B.1, fugitive emissions from solid
fuels activity linked to increased coal mining.

In Indonesia the three global datasets agree well up until 2010. From 2010, the inventories show a

continued increase in emissions, while the UNFCCC BUR emissions suggest a decline. EDGARv7.0 reports a
large increase in emissions from fugitive emissions from solid fuels (coal mining) (IPCC 2006, sector 1.B.1.) at
an increased average rate of 19 % per year and has increased by a factor of 152 until 2021 compared to 1990
(Figure 3).

For Russia, GAINS emissions are much higher than NGHGIs and the other two data sets due to the

revisions of the assumptions on the average composition of the associated gas generated from oil production
based on information provided in Huang et al. (2015). The higher emissions in GAINSv4 might be caused by a
greater source from venting of associated gas instead of flaring. GAINSv4 estimates a decline in global $CH_4$
emissions in the first half of the 1990s, primarily a consequence of the collapse of the Soviet Union and the
associated general decline in production levels in agriculture and fossil fuels (see regional emission illustrations
in figures S2–1 of the SI). In addition, as described by Evans and Roshchanka (2014) and assumed in Höglund-
Isaksson (2017), venting of associated petroleum gas declined significantly in Russia due to an increase in flaring.
It is unclear why this happened, but a possible explanation could be that the privatization of oil production in
this period meant that the new private owners were less willing to take the security risks of venting and invested
in flaring devices to avoid potential production disruptions. This hypothesis is however yet to be confirmed
(Höglund-Isaksson et al., 2020). FAOSTAT data for the Russian Federation starts in 1992, since the country did
not exist before this date. The former USSR statistics were used prior to 1992 without adjustments and this is the
cause of the 1990 and 1991 outliers in time series.. The slightly increasing trend observed in EDGARv7.0 and
FAOSTAT are set by emissions from the Energy sector.

For DR Congo estimates from GAINS are not available because they only report aggregated emissions

from a few African regions. Both FAOSTAT (PRIMAP based) and EDGARv7.0 estimates show similar slowly
increasing trends, potentially indicating the use of similar prior statistics (EFs). For non-AFOLU sectors the
PRIMAP-hist third party data priority scenario used in FAOSTAT also uses EDGAR data as an input data source
explaining similarities in these sectors. On the other hand, UNFCCC BUR data reports a strong increase in
emissions, which is due to a rapid growth of $CH_4$ emissions from the Waste sector, by a factor of four until 2018
compared to 2000. This increase happened at an average yearly rate of +8 %, with an initial sharp increase of
+30 % between 2000 and 2001. As previously discussed (section 3.1.) we believe that DR Congo BUR reported
Waste emissions are improbable and further investigation is needed.

For India, all bottom-up global inventories show similar trends and magnitudes of anthropogenic $CH_4$

emissions. The emissions of $CH_4$ averaged across EDGARv7.0, GAINS and FAOSTAT are 67% (2010), 68 %
(2014) and 65 % (2016) higher than the Indian BURs. All three BU inventories show an averaged steady increase
of 1 % yr$^{-1}$ between 1990-2020.

**3.3. NGHGIs compared to TD atmospheric-based $CH_4$ estimates**

Figure 4 presents the total TD estimates versus UNFCCC official reported emissions for the EU and the



seven non-EU emitters. The mean column on the right of each chart represents the mean of the overlapping time
series (2009-last available year, except for TROPOMI, which was available only for 2018-2020). For China, the
last BUR is available for 2014, and therefore we used that value. The inversions show total $CH_4$ emissions,
including both anthropogenic and natural sources. We present here the total TD estimate against the
anthropogenic NGHGIs, emphasizing that the difference between BU and TD estimates might be due to the
natural emissions.

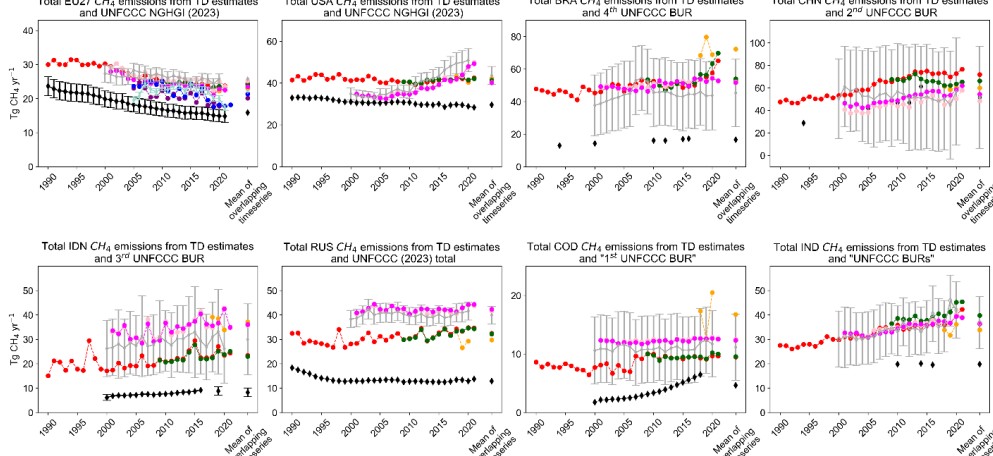

*Figure 4: Total anthropogenic $CH_4$ emissions (incl. LULUCF) from UNFCCC NGHGI (2023) CRFs (EU, USA*
*and Russia) and BURs (Brazil (4th in 2021), China (2nd in 2019), Indonesia (3rd in 2021), DR Congo (1st in 2022),*
*India (all three BURs: 2016, 2018 and 2021) and total TD estimates as following: for EU regional inversions*
*(FLEXkF_v2023, CIF-FLEXPART and CIF-CHIMERE) and global inversions (TM5-4DVAR,*
*CAMSv21r1_NOAA, CAMSv21r1_NOAA_GOSAT, CTE-GCP2021 and MIROC4-ACTM both runs) products.*
*The relative error on the UNFCCC value represents the NGHGI (2023) reported uncertainties computed with*
*the error propagation method (95% confidence interval) and gap-filled to provide respective estimates for each*
*year. China reports uncertainties for 2014 (min 5.2 %, max 5.3 %) and Indonesia reports for 2000 and 2019 ,*
*20 % and 19.9 % respectively. Total COD UNFCCC BUR emissions do not include IPPU. The last available*
*years are CIF-CHIMERE (2022), TM5-4DVAR, CIF-FLEXPART and CTE-GCP2021 (2020) and*
*FLEXkF_v2023, MIROC4-ACTM both runs, UNFCCC CRFs, and CAMSv21r1 both runs (2021). The mean of*
*overlapping time series was calculated for 2009-2021, except for TM5-4DVAR (2018-2020).*

Chandra et al., 2021 identify a few main sectors that triggered increases and decreases in the
anthropogenic $CH_4$ emissions of different countries. The first is Energy, fugitive emissions from the oil and gas
industry which helped to stabilize $CH_4$ concentration in the 1990s (decreased emissions), then they contributed
to the renewed $CH_4$ growth since the late 2000s (increased emissions). The other major sectors that drove changes



in the $CH_4$ growth rate arose from Agriculture (increase in emissions from enteric fermentation and manure
management) and from Waste. The increase in emissions from enteric fermentation and manure management is
caused primarily by increased animal numbers (AD), and in addition by the greater intensity of ruminant farming
as estimated by the FAO and the emission inventories (e.g. EDGAR) which might take into account productivity
increases (Crippa et al., 2020; Wolf et al., 2017; FAOSTAT, 2018) while inventory emissions from Waste can
account for up to 43 % of the linear increase in emissions for the rest of the world.
In the EU, the averaged 2009-2021 total $CH_4$ emissions from global inversions are in the range of 23-
26 Tg $CH_4$ $yr^{-1}$, in line with previous estimates published in Petrescu et al. (2021, 2023). As this is the total flux,
while the UNFCCC NGHGI (2023) report only anthropogenic emissions ($15.8 \pm 1.8$ Tg $CH_4$ $yr^{-1}$), the difference
can at least in part be explained by the sum of the natural emissions ( ~7 Tg $CH_4$ $yr^{-1}$). There is good agreement
in trends, but with inversions showing higher estimates. Tendentially, for the same period, regional inversions
report emissions between 20-22 Tg $CH_4$ $yr^{-1}$, lower than that of global inversions. Regional inversions use more
regional observations (e.g. ICOS, not just NOAA), have higher spatial resolution, and may thus better resolve
the transport. However, they may also have problems with the regional boundary conditions.
For the USA, averaged over the period 2009-2021, inversions indicate total $CH_4$ emissions in the range
of 40 - 44 Tg $CH_4$ $yr^{-1}$. In contrast, the UNFCCC NGHGIs (2023) report for the same period anthropogenic total
emissions of only 29 Tg $CH_4$ $yr^{-1}$. Regarding trends, those observed in the TD products are slightly increasing,
except for CAMS which shows no trend (Figure 4). The striking discrepancy between the trends from CAMS
and those from MIROC4-ACTM and CTE-GCP2021 are most likely caused by the increasing oil and gas
emissions from the Eastern USA (Permian Basin). The same increasing trend is also captured by GAINS (Figure
3). In their runs, both MIROC4-ACTM and CTE-GCP2021 use oil and gas priors from GAINS, while CAMS
uses priors from EDGAR (Figure 3). In SI, we discuss further differences in having CTE-GCP2021 run with
both EDGAR and GAINS oil & gas prior estimates.
For Brazil, inversions yield an average (range) total $CH_4$ emissions of 55 (42-72) Tg $CH_4$ $yr^{-1}$, with
TM5-4DVAR reporting the highest estimate. The UNFCCC estimate of anthropogenic emissions is 16.6 Tg $CH_4$
$yr^{-1}$. The two CAMS inversions report an increased trend during 2017-2021, with 15 Tg $CH_4$ higher emissions
in 2021 than in 2017. There is also a peak in the TROPOMI observation in 2019, and the TM5-4DVAR model
attributed this to biomass burning events, identified in the reported partitions (see
https://doi.org/10.5281/zenodo.10276087 data figure).
For China, approximately 80 % of the $CH_4$ emission increase (21.5 Tg $yr^{-1}$) during 2000 – 2015 was
from fugitive emissions from coal (mines), consistent with what GAINS and EDGAR reports (Figure 3). The
TD estimates mostly agree, except for CAMS inversions which find 10 to 20 Tg $CH_4$ $yr^{-1}$ higher emission than
the other inversions. Both MIROC4-ACTM runs (control and OH inter-annual variability (IAV) varying run;
Patra et al., 2021) are in line with the BURs. Trend wise, all inversions agree on increased emissions after 2019.
Nevertheless, CAMS_GOSAT_NOAA registers a decrease after 2013 not seen in the other inversion trends.
For Indonesia, most TD results agree on the trend and show a slight increase in emissions. Similar trend
is also seen by the BURs. However, the CAMS inversions result in about 10 Tg $CH_4$ $yr^{-1}$ lower emissions than
the other inversions (MIROC4-ACTM and CTE-GCP2021). Regarding the East Asian estimates,
MIROC4_ACTM inversion simulates higher fluxes compared to the other inversions. Only recently they found



that annual total East Asian emissions have lowered more significantly than in Patra et al. (2016) or Chandra et
al. (2021).
For Russia, the estimates from CAMS are both in the same range as the BU GAINS estimate (see Figure
2) from 2000 onwards (between 30-40 Tg CH$_4$ yr$^{-1}$) but does not show such a strong decrease as GAINS from
1990 to 2000, Fig. 2), while MIROC4-ACTM and CTE-GCP2021 are about 10 Tg CH$_4$ yr$^{-1}$ higher than CAMS).
For DR Congo, except for the TM5-4DVAR which also reports the highest emissions (wetlands - see
partitions, Fig.5a), inversions do not show any trend. MIROC4-ACTM runs report about 5 Tg CH$_4$ yr$^{-1}$ higher
emissions than both CAMS runs. The two high values in 2018 and 2020 seen by the TROPOMI satellite are
triggered by high emissions from wetlands reported in the TM5-4DVAR partitions.
For India, the TD estimates of total emissions agree well on increased trends and magnitudes. In
contrast, UNFCCC reporting does not show a trend, but too little reported data from BURs is available, therefore
a plausible conclusion cannot be drawn.

**3.4. Reconciliation and sectoral attribution of CH$_4$ emissions**
***3.4.1. Sectoral attribution of CH$_4$ emissions in TD products***

Table 2 shows the partitions as originally reported by some of the inversions, which we name here
"unharmonized partitions". A straightforward, direct comparison of the fluxes is not possible because of the
different ways each inversion allocates and groups the natural/anthropogenic fluxes. For example, not all
inversions report soil fluxes as done by MIROC4-ACTM and CTE-GCP2021 (together with wetlands), or report
the biomass burning fluxes separately from anthropogenic emissions (MIROC4-ACTM and TM5-4DVAR).
Also the termites, oceans and geological fluxes are sometimes reported separately (MIROC4-ACTM) or grouped
in "Other" (CTE-GCP2021, TM5-4DVAR). Regarding the anthropogenic emissions, TM5-4DVAR reports them
as other, providing a separate partition for rice.

*Table 2: Unharmonized partitions originally reported by inverse products:*

| Inversion | Anthropogenic | Rice | Soils | Wetlands | Ocean | Termites | Geological | Biomass burning | Other |
|---|---|---|---|---|---|---|---|---|---|
| **CAMSv21r1 (both runs)** | Yes (in Other) | Yes | No | Yes | Yes (in Other) | Yes (in Other) | No | Yes | Yes |
| **MIROC4-ACTM (control and OH var)** | Yes ((Agr, Waste, Oil/Gas, Biofuel, coal) | Yes (in Agr.) | Yes | Yes | Yes | Yes | Yes | Yes | Yes (separated) |
| **CTE-GCP2021\*** | Yes (Agr, waste, fossil fuel, biofuel, biomass burning) | Yes (in Agr.) | Yes (BIO) | | Yes (In Other) | Yes (In Other) | Yes (In Other) | In anthr. | Yes (Ocean, Termites, Geological) |
| **CEOS (GOSAT)** | Yes (Livestock, rice, waste, coal, oil, fire) | In anthr. | No | Yes | No | No | Yes (seeps) | In anthr.(but separate) | only seeps |
| **TM5-4DVAR (TROPOMI)** | Yes (in Other) | Yes | No | Yes | Yes (in other) | Yes (in other) | Yes (In Other) | Yes | Yes\*\* |



| Inversion | Anthropogenic | Rice | Soils | Wetlands | Ocean | Termites | Geological | Biomass burning | Other |
|---|---|---|---|---|---|---|---|---|---|
| **GEOS-Chem CTM (TROPOMI for USA)** | Yes (Livestock, Oil Gas, Landfills, Wastewater, Other anthro (rice) | In. other anthr. | No | Yes | Yes (In Other) | Yes (In Other) | Yes (In Other) | Yes (In Other) | Yes*** |

*CTE-GCP2021 partitions refer to anthropogenic, bio and other.
** In TM5-4DVAR (similar to the CAMSv20 set-up and CAMSv21r1), the "Other" partition includes anthropogenic sources
except for the rice paddies. It also includes the small fluxes from termites, oceans, soil sink, geological etc.). More details on
priors are found in Petrescu et al., 2023b, Priors table.
***Named Other biogenic

Figure 5 shows the UNFCCC NGHGI anthropogenic total reported estimate (diamond) next to all TD
estimates. All global inversions report total and dissagregated partitions, while the regional inversions report
only the total emissions (green column).

UNFCCC anthropogenic and unharmonized $CH_4$ emissions from partitions reported by TD estimates (average 2015-last available year)

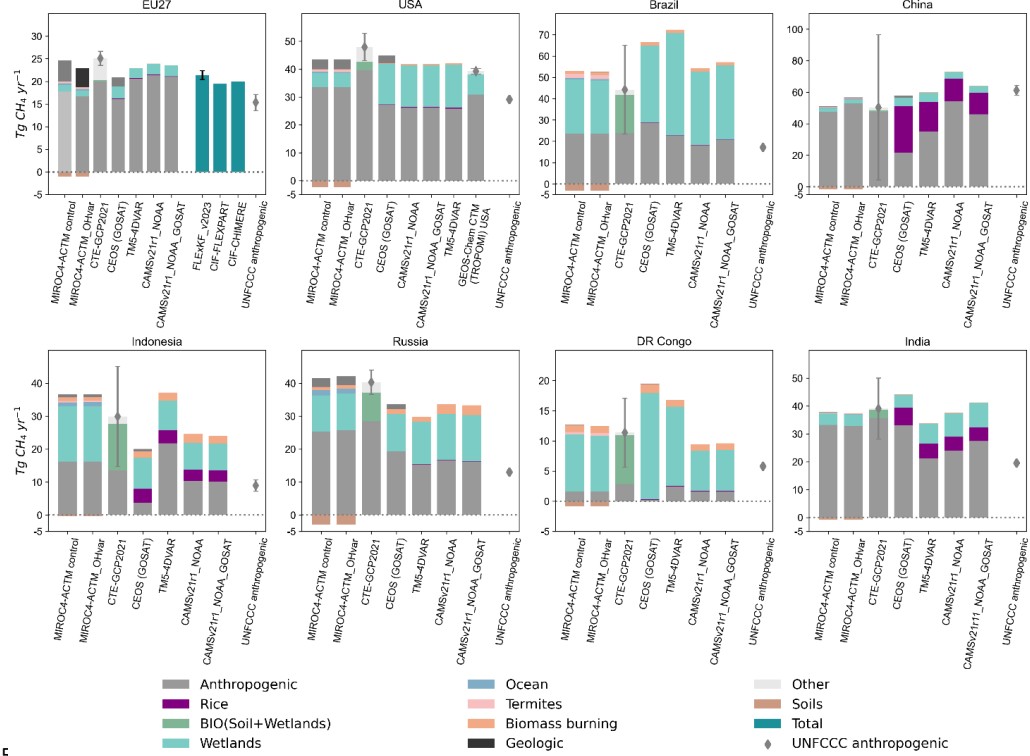

*Figure 5: Total (green) and disaggregated anthropogenic and natural CH$_4$ emissions from TD estimates*
*compared to UNFCCC NGHGI (2023) anthropogenic emissions (incl. LULUCF) (diamond) for the EU and*





*seven global emitters outside the EU (USA, Brazil, China, Indonesia, Russia, DR Congo and India). The*
*UNFCCC anthropogenic value represents the sum of all five IPCC sectors (Energy, IPPU, Agriculture,*
*LULUCF and Waste). The partitions reported by the TD global inversions are detailed in Table 2. The relative*
*error on the UNFCCC CRF value represents the NGHGI (2023) reported uncertainties computed with the error*
*propagation method (95% confidence interval) and gap-filled to provide respective estimates for each year (see*
*Petrescu et al., 2023a, Appendix). China value and uncertainties (min 5.2 %, max 5.3 %) are for 2014 only and*
*Indonesia uncertainties for 2019, 19.9 %. For the USA CEOS (GOSAT) we used the Nessar et al., 2023 total*
*uncertainty of min 1.1 and max 1 Tg yr$^{-1}$. CTE-GCP2021 provides uncertainties for each partition, but here the*
*uncertainty of the total flux is shown. FLEXkF_v2023 reports the relative uncertainty (%) of the posterior*
*emissions. The plotted data represents the average between 2015 and last available year as follows: CIF-*
*CHIMERE (2022), TM5-4DVAR, CIF-FLEXPART and CTE-GCP2021 (2020) and FLEXkF_v2023, MIROC4-*
*ACTM both runs, UNFCCC CRFs, and CAMSv21r1 both runs (2021). GEOS-Chem CTM (TROPOMI) USA*
*reports only for 2019 (Nesser et al., 2023).*
We note that CTE-GCP2021 reports the net natural land-biosphere flux "BIO flux" (soil+wetlands),
while other inversions report wetlands and soil separately, thus the BIO flux will be smaller than the wetlands,
because mineral soils are a CH4 sink. Regarding the allocation of different fluxes to partitions, sometimes Rice
emissions are part of the Agriculture component (anthropogenic partition) (MIROC4-ACTM, CTE-GCP2021)
while CEOS (GOSAT) and GEOS-Chem CTM (USA TROPOMI) report separate partitions for Rice in
Anthropogenic emissions. Same for the biomass burning - CTE-GCP2021and CEO report it as part of
Anthropogenic emissions, while GEOS-Chem CTM as part of Others. The rest of the inversions report it
separately; this different allocation makes comparisons for these two sources challenging.

To facilitate comparisons between all TD products, we aggregated and harmonized the partitions in
three main categories, as summarized in Table 3 and Figure 6. The dark green columns in Figure 6 show the total
flux for regional EU inversions which did not report partitions.

*Table 3: Harmonized partitions from inverse products:*

| Inversions | Anthropogenic + Rice + Biomass burning | | | Soils + Wetlands | | Other (Ocean + Termites + Geological) | | |
|---|---|---|---|---|---|---|---|---|
| | **Anthropogenic** | **Rice** | **Biomass burning** | **Soils** | **Wetlands** | **Ocean** | **Termites** | **Geological** |
| **CAMSv21r1 (both runs)** | = Other | Yes | Yes | No | Yes | Yes | Yes | Yes |
| **MIROC4-ACTM (control and OH var)** | Yes ((Agr (livestock + **rice**), Waste, Oil/Gas, Biofuel, coal) | In Agr. | Yes, summed to anthr. | Yes | Yes | Yes | Yes | Yes |
| **CTE-GCP2021\*** | Yes (Agr (**rice** is in), waste, fossil fuel, biofuel, biomass burning) | in Agr. | In anthr. | Yes (BIO) | | Yes (Other) | | |
| **CEOS (GOSAT)** | Yes (Livestock, rice, waste, coal, oil, fire) | In anthr. | In anthr. | No | Yes | No | No | Yes |



| | | | | | | | | |
|---|---|---|---|---|---|---|---|---|
| **TM5-4DVAR (TROPOMI)** | Others + Rice+ BB | In anthr. | Yes, summed to anthr. | In Other | Yes | Yes | Yes | Yes |
| **GEOS-Chem CTM (TROPOMI) USA** | Yes | In anthr. | In other biogenic | No | Yes | Yes | Yes | Yes |

*CTE-GCP2021 partitions refer to Anthropogenic, Bio and Other. Other fluxes are imposed

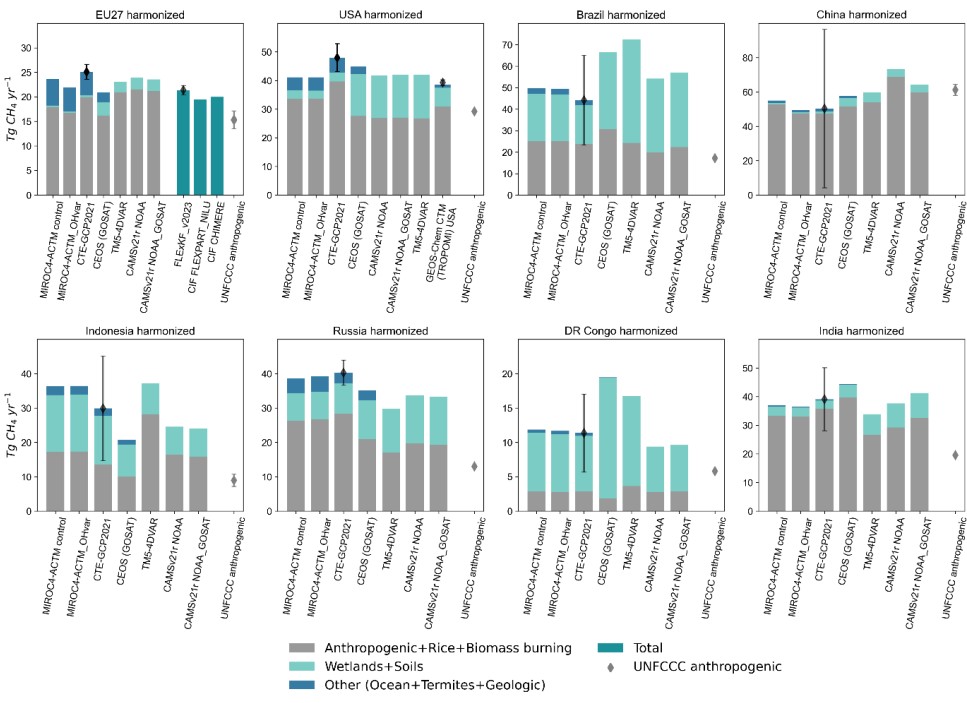


*Figure 6: Total (green) and disaggregated anthropogenic and natural $CH_4$ emissions from TD estimates compared to UNFCCC NGHGI (2023) anthropogenic emissions (incl. LULUCF) for the EU and seven global emitters (USA, Brazil, China, Indonesia, Russia and DR Congo). The UNFCCC anthropogenic value represents the sum of all five IPCC sectors (Energy, IPPU, Agriculture, LULUCF and Waste). The partitions reported by the TD global inversions are harmonized and detailed in Table 3. The relative error on the UNFCCC CRF value represents the NGHGI (2023) reported uncertainties computed with the error propagation method (95% confidence interval) and gap-filled to provide respective estimates for each year (see Petrescu et al., 2023a, Appendix). In 2014, China UNFCCC value and reported uncertainties (min 5.2 % and max 5.3 %) are for 2014 while Indonesia reported uncertainties for 2019, 19.9 %. India UNFCCC value is for 2016. CTE-GCP2021 provides uncertainties for each partition, but here we plotted the uncertainty of the total flux. FLEXkF_v2023 reports the relative uncertainty (%) of the posterior emissions. The plotted data represents the average between 2015 and last available reported year as follows: CIF-CHIMERE (2022), UNFCCC CRFs, TM5-4DVAR, CIF-*



*FLEXPART and CTE-GCP2021 (2020) and FLEXkF_v2023, MIROC4-ACTM both runs, and CAMSv21r1 both*
*runs (2021). GEOS-Chem CTM (TROPOMI) USA reports only for 2019 (Nesser et al., 2023).*
**3.4.2. Reconciliation of BU and TD CH$_4$ estimates**

Figure 7 summarizes the total CH$_4$ fluxes for the EU and the seven global emitters as following: BU

anthropogenic sources disaggregated per sectors, BU natural emissions, TD natural emissions from regional and
global inversions, and total emissions from global TD estimates (see 2.3 and SI for description of all data
products).

The way data is currently reported by the inversions, is inconsistent regarding the comparison between

BU natural and TD natural sources. TD products differ in the sources they report (Table 2) or they allocate them
to different categories. We consider natural the following sources: biomass burning, soils, oceans and termites
(often reported by inversions under category "Other"), wetlands, geological and lakes & reservoirs (or
freshwaters). Due to lack of information, biomass burning emissions were considered among the natural sources,
recognizing that in regions like tropical forests, some of these events are influenced by human intervention. To
make the products from Figure 7 comparable, we added the missing BU information from TD, and vice-versa,
presented in hashed pattern. In this way, comparison between BU and TD natural emission estimates is consistent
regarding the "apples to apples" comparison, but became "apples of different flavors" (see Table 4):

*Table 4: BU and TD natural partitions as presented in Figure 7:*

| Product name | TD natural partitions | | |
|---|---|---|---|
| | reported | missing* | added from |
| **TM5-4DVAR (TROPOMI)** | BB and wetlands | oceans, termites, soils, geological, lakes and reservoirs | MIROC4-ACTM (termites, oceans and soils), DAAC lakes and reservoirs, geological, updated for this study (SI) |
| **CEOS (GOSAT)** | Fires (BB), Seeps and wetlands | termites, oceans, soils, lakes and reservoirs | MIROC4-ACTM (termites, oceans and soils), DAAC lakes and reservoirs |
| **MIROC4-ACTM control** | BB, wetlands, oceans, termites, soils, geological | lakes and reservoirs | DAAC lakes and reservoirs |
| **MIROC4-ACTM_OHvar** | BB, wetlands, oceans, termites, soils, geological | lakes and reservoirs | DAAC lakes and reservoirs |
| **CAMSv21r1_NOAA** | BB, wetlands, "Others" include anthropogenic and was not used | termites, oceans, soils, lakes and reservoirs, geological | MIROC4-ACTM (termites, oceans and soils), DAAC lakes and reservoirs, geological, updated for this study (SI) |
| **CAMSv21r1_NOAA_GOSAT** | BB, wetlands, "Others" include anthropogenic and was not used | termites, oceans, soils, lakes and reservoirs, geological | MIROC4-ACTM (termites, oceans and soils), DAAC lakes and reservoirs, geological, updated for this study (SI) |
| **CTE-GCP2021** | soils + wetlands (BIO), termites and oceans | BB | BB from GFEDv4.1s |
| **Product name** | BU natural partitions | | |
| | available | missing** | added from |



| GFEDv4.1s DAAC LPJ-GUESS Geological | GFEDv4.1s DAAC LPJ-GUESS Geological updated in this study (SI) | soils termites oceans | MIROC4-ACTM |
|---|---|---|---|

note: in TD products termites, oceans emissions are imposed from existing literature
* presented as hatched pattern in the figure "\\\"
** presented as hatched pattern in the figure "///"

We note from Figure 7 that in all Annex-I countries (EU, USA, Russia) and China, TD and BU natural
emissions are consistent with each other, after including the missing sources, as detailed in Table 4. For Brazil
and DR Congo, the gap between the two natural components is highly significant, while less for Indonesia and
India. We hypothesize that mapping of the wetlands extent might cause these inconsistencies.
For an easier visual comparison and reconciliation between BU and TD estimates, we added in white
the mean of the BU anthropogenic estimates, underneath the BU and TD natural estimates. We note that for most
countries, the sum of the anthropogenic and natural components matches those of the TD global total estimates.
This gives confidence that, to a certain extent and albeit inconsistencies between products, BU anthropogenic
emission estimates are accurate and consistent with the observation-based estimates and can be used to reconcile
with the atmospheric-based estimates.

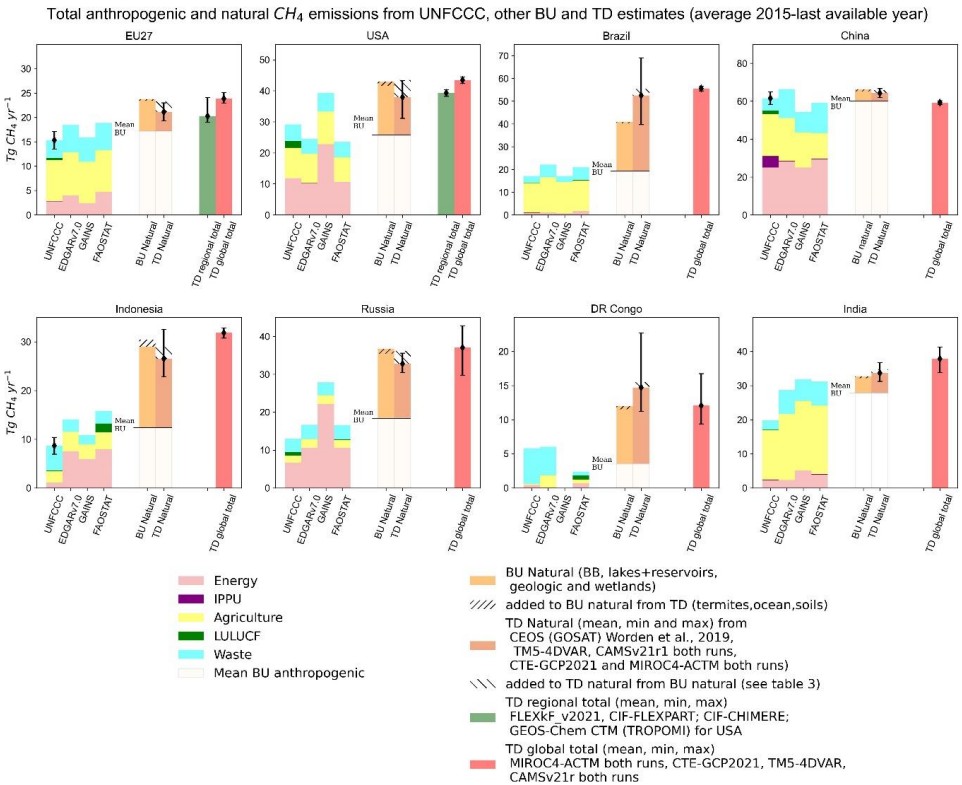

62

*Figure 7: Total anthropogenic and natural CH₄ emissions from BU and TD estimates presented as average of*

*2015-last available year for EU and seven global emitters (USA, Brazil, China, Indonesia, Russia, DR Congo*

*and India). The BU anthropogenic estimates belong to: UNFCCC NGHGI (2023) CRFs and BURs (incl.*

*LULUCF) as totals and sectoral shares, EDGARv7.0, GAINS and FAOSTAT-PRIMAP. The relative error on the*

*UNFCCC CRF value represents the NGHGI (2023) reported uncertainties computed with the error propagation*

*method (95% confidence interval) and gap-filled to provide respective estimates for each year (see Petrescu et*

*al., 2023a, Appendix). In 2014, China reported an uncertainty of min 5.2% - max 5.3%. The BU Natural*

*emissions for the EU are the sum of the VERIFY products (biomass burning, inland waters, geological and*

*peatlands plus mineral soils as described in Petrescu et al., 2021 and 2023a, Appendix A2.1). For the seven non-*

*EU emitters, the BU Natural fluxes are the sum of wetland emissions (LPJ-GUESS), lakes and reservoirs fluxes*

*(ORNL DAAC, Johnson et al., 2022), geological (updated activity in SI) and biomass burning emissions*

*(GFED4.1s). The TD natural global estimates are presented in Table 1. The uncertainty on the TD natural*

*emissions is the min/max of all estimates. To both BU and TD estimates missing information was added (see*

*Table 4). The natural emissions have been plotted starting at the mean of the BU anthropogenic estimates, to*

*retain comparability across the natural emission estimates, but also compare with the total TD estimates. The*

*total regional TD estimates (for EU) belong to the mean and min/max of FELXkF_v2023, CIF-FLEXPART and*

*CIF-CHIMERE and for USA GEOS-Chem CTM (TROPOMI) for the year 2019 (Nesser et al., 2023). The total*

*global TD inversions represent the average of the 2015-last available year of the mean and min/max of CTE-*



*GCP2021, MIROC4-ACTM both runs, CAMS v21r both runs and TM5-4DVAR. The last available years are*
*2022 for CIF-CHIMERE, 2021 for EDGARv7.0, FAOSTAT, MIROC4-ACTM both runs, UNFCCC CRFs, and*
*CAMSv21r1 both runs, and 2020 for CIF-FLEXPART and CTE-GCP2021. TM5-4DVAR partitioned data is only*
*available between 2018-2020.*

However, this should be interpreted with caution because in Europe, natural emission priors come from regional ecosystem model simulations, where drained peatland, drainage ditches areas, and pristine areas are lumped together. Therefore, if both LULUCF sector and natural BU emissions are included in the total budget estimation, there is some overlap and possible double-counting. Especially, ecosystem model estimates of 'soil sink' or 'inundated soil emissions' may be overlapping with NGHGI managed peatland forest soil category (or agricultural soils). The separation of emissions into different categories requires further clarification together with inventory makers. Furthermore, it should be assessed which emissions should be called natural and which anthropogenic (e.g., LULUCF, Agriculture) by inversions.

## 4. Discussion and recommendations on reconciliation procedures

Figures 3, 4a, b, 5 and 6 visually compare inversions and inventory estimates. A valid comparison should include consistent types of information (tiers, priors, methodology) and show a full uncertainty analysis to determine whether differences between estimates are statistically significant. However, very few datasets provide this necessary uncertainty information that is critical for interpreting comparisons. Furthermore, from the perspective of verification, some adjustment of estimates (additions/subtractions) may be required to reflect variations in which anthropogenic and natural sources and sinks are included in (or excluded from) the respective estimates. The following paragraphs discuss the potential challenges and opportunities for comparing NGHGI against external BU and TD estimates from the perspective of national agencies responsible for inventory compilation and reporting.

The two most common issues were identified by the series of previous syntheses and other scientific literature (Petrescu et al., 2021, 2023; McGrath et al., 2023, Andrew 2020; Grassi et al., 2018). The first is the **geographical scope and boundaries** of inverse modelling versus inventory estimates. Independent of which GHG are analyzed, a general system boundary issue is masking of gridded results to the country level, where it is important to know how modelling groups have defined emissions in each grid cell and to ensure the mask correctly captures the territorial perspective (e.g., country and economic zones), in line with how official NGHGIs are reported (EEA, 2013). Given that NGHGIs do not report gridded emissions, comparison with gridded inversions may introduce some under/over estimation of total emissions within the country border. Particularly for inversion models with a coarser grid, aggregation of the grid cells to country levels will not necessarily give a perfect match. This problem lessens with larger regions, or regions with long coastlines, and is one reason why VERIFY (https://verify.lsce.ipsl.fr/) aggregates many smaller countries together to larger regions. The second issue linked to boundaries is the **structural system boundaries**, an inconsistency with great implications in comparing the inventory- with inversions-based estimates for **source attribution**, e.g., anthropogenic vs. natural. Most emission inventories aim at estimating anthropogenic emissions, while most inversions see both anthropogenic and natural emissions. Thus, methods are needed to separate the anthropogenic



flux from the total flux (Deng et al. 2022, and above section 3.4), similarly to what was done in Andrew 2020
who refers to system boundaries when comparing the various inventory datasets focusing on what sources are
included in the respective estimates. This is a particularly important issue for $CH_4$ where, globally, natural
emissions are of similar magnitude as anthropogenic emissions, with larger variations at regional scales, mainly
due to seasonality (i.e. wetlands). Further, climate change may modify natural emissions in ways that models
can't yet resolve, for example, a warmer climate may increase natural emissions of $CH_4$ (Yvon-Durocher., et al.,
2014).

Another identified source of inconsistencies between inventory estimates and inversions-based
estimates are **uncontrolled events**. $CH_4$ from the fossil fuel industry can contribute to high releases to the
atmosphere over a short period of time, given the large number of uncontrolled emission point sources in oil and
gas (O&G) and coal production areas worldwide (Jackson et al., 2020). Such processes include leakage from
landfills, spontaneous events from oil and gas production activities, so-called uncontrolled gas well blasts etc. In
recent years, there is a high interest in quantifying emissions from these events (Jacob et al., 2016) and a series
of top-down studies using satellite imagery quantifies these sparse but important events, which are difficult to
include in the national inventories leading to a potential underestimate of emissions. However, they can be
identified and quantified to ultimately be included in the inventories, as done for the USA by Massakkers et al.,
2016 and 2022. Recently, under the $CoCO_2$ project (https://coco2-project.eu/) a hot-spot satellite detection
interactive map (Published studies on hot spot detection (CO2, CH4) - uMap (openstreetmap.fr) was released as
a user-centric interface featuring published studies on hot-spot detection between 2010 to 2021. It allows for
advanced filtering by year, gas, activity, geographical zone, and country.
Varying **temporal and spatial resolutions** are also an important factor to consider. Inventory-based
estimates use annual temporal resolutions and are not spatially distributed yet are compiled at a high source-
sector resolution. Inversion models can produce estimates at a potentially fine grid scale (kilometers) and fine
temporal resolution (hours). The optimized net fluxes and/or partitioned anthropogenic and natural flux totals
may be too coarse to identify biases at the source-sector level used in inventories. A region (e.g. city or region
within a country) would be interested in annual emissions, but the availability of more detail could help identify
and manage 'events' (acute pipeline leaks) from individual facilities (power plant or industrial site), and this may
help inventory agencies verify emissions from these facilities. The fine spatial resolution allows aggregation to
city- or region-level, matching as close as possible to jurisdiction boundaries. Uncertainties in inversion estimates
reduce when aggregating in time and space since the grid-cell errors are often negatively correlated. Thus, from
TD, we can be more confident in estimates for large regions than in estimates for small regions or for a single
grid cell. This problem lessens with denser observational coverage.
When comparing inventory- and inversion-based emissions, there are difficulties in analyzing **trends**
due to different time scale **variability**. Inventory-based approaches report emissions at the annual level, but often
do not consider intra-annual (seasonal) variations, important for the microbial sources. Further, the Paris
Agreement is set around five-yearly Global Stocktakes, which indicates a desire to average trends, prioritizing
the multi-annual trend over IAV, canceling out extremes from both weather and socio-economic fluctuations.
Inversion models, on the other hand, include variations over a wide range of timescales, but in particular for IAV
(e.g. OH and weather) that remains challenging to assess. For an effective comparison, inversion-based estimates



need to have IAVs statistically removed to make comparisons with NGHGIs easier (e.g., 5-year or 10-year
averages or trend analysis). Additionally, averages of ensembles of inversions may mask underlying differences
and trends in individual inversions. More broadly, methods to identify if a difference between two independent
datasets is statistically significant (levels of trends) exist but are inhibited by: a) information on inventory
uncertainty; and b) computation cost to generate inversion emission uncertainties. A suite of approaches to
resolve uncertainties in both BU and TD methods, and to enable comparisons between approaches, will be
important for the design and usefulness of future integrated monitoring systems. ome attempts have been initiated
(e.g. CIF (Berchet et al., 2021) in the VERIFY H2020 project and the $CH_4$ inversion intercomparison funded by
WMO).
From the point of view of the observation information, a more valid comparison between inversions is
made when all inversions use the same **priors**. In this context, we define as priors input data in the form of
atmospheric observations (e.g. satellite retreivals, ground-based observation networks (ICOS)) and/or bottom-
up emissions datasets (e.g. EDGAR, GAINS) used as input parameters to the inverse models. When combined
with observation data, the inversion system produces a posterior estimate of emissions, which can then be
compared to the prior estimate, preferably incorporating a full uncertainty analysis. The posterior emissions
depend to some extent on the prior that was used; the extent of this dependency is determined by the number of
observations used in the inversion, by how the observations relate to the emissions (governed by atmospheric
transport) and by the uncertainties assigned on the prior emissions and the observations. Thus, the inversions
would be more robust with better quantified uncertainties for the prior emissions. Whereas the comparison on an
inversion with NGHGIs or other inversions would be made more robust by having more information on how
dependent the posterior estimate is on the prior. This stresses the need for more systematic measurements of
fluxes necessary to produce adequate prior data (Bastviken et al., 2022) and synthesized atmospheric
observations with their uncertainties to robustly constrain the inversions.
It is not generally clear how inventory uncertainties can be compared to inversion uncertainties;
however, it is important that both methods provide comprehensive **uncertainty estimates**. The "uncertainty
reduction" characterizes the improvement brought by the inversion over the prior emission estimate. The prior
emissions used as input into an inversion model should have uncertainty statistics, and a full inversion analysis
will include uncertainties on the posterior estimate, with the reduction in uncertainty between the two estimates
of particular interest. The inventory-based emission estimate will additionally have uncertainty estimates, though
some argue these statistics are not sufficiently robust for verification purposes (National Academies of Sciences,
Engineering, and Medicine 2022). There are often offsets in inversion models, because of systematic
inconsistencies between observations and chemistry-transport models, which may make trends more robust than
instantaneous estimates. Strongly dependent on the use of the information, uncertainties might be given different
weights, e.g. in a policy context, the uncertainty on the emission trend may be more important than emission
level uncertainties. However, this is harder to estimate as it requires knowledge of correlations in emission
estimates over time. Given the above listed challenges in quantifying uncertainties, research projects such as
VERIFY and global initiatives like the Global Carbon Project (GCP), make use of multi-model analysis
(**ensembles**). This way the UNFCCC emission inventory is compared against, for example, 13 land surface
models and 22 inversion models (Saunois et al., 2020; Deng et al. 2022). From a scientific perspective, the model





ensemble is often considered a more robust estimate of the mean and uncertainty, as transport model uncertainty can be accounted for. From an inventory perspective, individual model comparisons may be more efficient, as various input variables or processes can be compared directly to the inventory. Doing this for each model becomes time consuming. Currently, most inventory comparisons in UNFCCC National Inventory Reports (UK, Switzerland) use single model comparisons.

Finally, there are no **standardized procedures** for inverse analysis systems; therefore, there is room for additional progress and refinement of emission estimates and uncertainties derived from atmospheric observation and inverse models. The Community Inversion Framework (CIF; Berchet et al., 2021) is a good example and moves in this direction. However, improvements are still needed to ensure common formatting and presentation of the results, in addition to the use of common language and terminology.

Based on the above discussions, improvements/additions in the following areas could enhance the information that national inventory agencies could gain from comparing NGHGIs with other BU and TD estimates:

1. Generate spatially distributed data of NGHGIs by the respective inventory agencies.
2. A better quantification of uncontrolled spontaneous events of point source estimates to complement NGHGIs (Maasakkers et al., 2016 and 2022 for the EPA in the USA).
3. Denser atmospheric observation networks to feed into inversions and reduce uncertainties at grid-cell levels.
4. Routine provision to quantify uncertainties from inversion models.
5. Develop methods to compare estimates and ensembles with inventory estimates.
6. Increase quality in systematic measurements of fluxes to produce better priors with better uncertainties.
7. More accurate transport models to increase robustness in emission estimates.
8. Clear prescribed inventory methods to estimate and assess uncertainties, particularly statistical significance.
9. Accurate model outputs (EFs) to be used in direct comparisons with inventory data.
10. Accurate estimates of natural emissions with better spatial distribution, to subtract from total inversion estimates and compare with anthropogenic NGHGIs.
11. Develop a common language, terminology and data formats to compare inversions with NGHGI formats.

## 5. Data availability

Data files reported in this work which were used for calculations and figures are available for public download at https://doi.org/10.5281/zenodo.10276087 (Petrescu et al., 2023b). The data are reachable with one click (without the need for entering login and password), with a second click to download the data, consistent with the two click access principle for data published in ESSD (Carlson and Oda, 2018). The data and the DOI number are subject to future updates and only refer to this version of the paper. The raw gridded data is available upon request, directly from the data providers, as detailed in the Supplementary Information, Table S2.

## 6. Conclusions

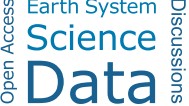

We analyzed data from both anthropogenic and natural CH$_4$ fluxes, from both BU and TD observation-
based estimates (Table 1). BU estimates show that, for most non-Annex I countries, the largest CH$_4$ emissions
are from the Energy sector, followed by emissions from the Agricultural sector (Figure 7), primarily from
ruminant livestock (enteric fermentation). The inversions (Figure 5) attribute most of the fluxes to the
anthropogenic emissions (gray) including rice (purple) (EU, China) while in tropical countries emissions are
attributed mostly to natural processes (wetlands).
The EU and the seven large emitters analyzed here contribute an anthropogenic emission of 173 Tg CH$_4$
yr$^{-1}$ (sum of last UNFCCC reported year, Figure 1,2), representing roughly half of the total global anthropogenic
emissions (386 Tg CH$_4$ yr$^{-1}$) reported by EDGARv7.0 in 2021, while the average of the anthropogenic component
from the atmospheric global inversions (MIROC booth runs, CTE-GCP2021, CEOS and CAMS booth runs) is
181 Tg CH$_4$ yr$^{-1}$ (Figure 5). Regarding comparability issues (Petrescu et al., 2023b, Matrix products table),
comparisons between UNFCCC and BU products (Figure 3) highlight substantial deviations, likely mainly due
to assumptions regarding gas/oil emissions (e.g., GAINS for Russia and the USA).
We highlighted the challenge of reconciling BU and TD estimates, due to different priors used in the
simulations (Petrescu et al., 2023b, Priors Table). It is also challenging to compare different TD products due to
the source attributions and allocation of fluxes to different sectors and activities within sectors (Table 2 and
Figure 5).
The comparison between UNFCCC and the TD estimates (Figure 4) agrees largely with the findings of
Deng et al. (2022) who applied different methodologies to calculate natural emissions. This approach produced
different results and highlighted the need for better estimation of natural fluxes from ecosystem-based models,
similar to our findings (Brazil and DR Congo, Figure 7). For the moment, it is difficult to discuss and draw
conclusions regarding emissions trends seen by inversions. From Figure 4, only UNFCCC shows clear patterns
during the last three decades (e.g. declined in the USA, and EU (regulations) and Russia (dissolution of the Soviet
Union) because are driven mainly by the anthropogenic component which is better constrained in bottom-up
inventories, while inversions include the natural component as well. Therefore, as described above in section 4,
comparability issues such as IAV and seasonal variability might strongly influence trends.
Given that, in most cases, the UNFCCC BURs reports are incomplete for the non-Annex I parties
(China, Indonesia, DR Congo) it is important to acknowledge that the TD estimates might become a useful way
to complement inventories and play a role in the validation of the BU estimates. In most cases, the gap between
the anthropogenic BU fluxes from inventories and total TD fluxes can be largely explained by the natural fluxes
(Figure 7).
There is still a pressing need for reporting of uncertainties in both prior and posterior emissions, even if
some TD inversions do report it (CTE-GCP2021 and FLEXkF_v2023, Figure 4) as the standard deviation of
ensemble members. The use of a variety of priors across different inversion systems can also inhibit
comparability with inventories and between inversions. Generally, inversions are still ill-constrained by
observations (only 60 sites globally plus satellites) and the prior flux uncertainty for each of the 54 regions is
large. Therefore, the monthly results could be more ill-constrained than the annual totals.
Even if comparisons between CH$_4$ inversion estimates and NGHGIs are currently uncertain because of
the spread in the inversion results, TD inversions inferred from atmospheric observations represent independent



data against which inventory totals and trends can be compared, considering the recommendations listed at the
end of section 4.

## 7. Appendix

All the information regarding models/methods descriptions is available in the Supplementary Information (SI)
file. Appendices A1 and A2 in Petrescu et al., 2023a contain detailed information about Table 1 products. Further
information on new products together with references and contact details are found in Tables S1 and S2 in SI.

The tables with priors used by all the products and the matrix highlighting the comparability issues identified in
section 4 are found in the Zenodo data repository, Petrescu et al., 2023b.

## Supplementary Information (link)

## Author contributions

AMRP designed research and led the discussions; AMRP wrote the initial draft of the paper and edited all the
following versions; GPP drafted the initial version of section 4, edited the final version of this manuscript and
advised on the context; PP processed all the original EU data submitted to the VERIFY portal; RLT, SH, BM,
DaB, RL, PKP, AT, RMA, LHI, FNT, GC and JG edited and gave consistent comments and suggestions to the
manuscript; all co-authors are data providers and contributed to subsequent versions of the manuscript by
providing specific comments and information related to their data in the main text, providing as well product
descriptions for the Supplementary Information file.

## Competing interests

At least one of the (co-)authors is a member of the editorial board of Earth System Science Data

## Acknowledgements

The lead author would like to thank former colleagues, Dr. Chunjing Qiu and Dr. Matthew McGrath for previous
work done in the VERIFY project. FAOSTAT statistics are produced and disseminated with the support of its
member countries to the FAO regular budget. The views expressed in this publication are those of the author(s)
and do not necessarily reflect the views or policies of FAO. We acknowledge the work of current and former
members of the EDGAR group (Marilena Muntean, Diego Guizzardi, Monica Crippa, Edwin Schaaf, Efisio
Solazzo, Gabriel David Orreggioni and Jos Olivier).

## Financial support

This research has been supported by the European Commission, Horizon 2020 Framework Programme (CoCO2,
grant no. 958927).



Development of MIROC4-ACTM is supported by the Environment Research and Technology Development
Fund (grant no. JP-MEERF21S20800) and the Arctic Challenge for Sustainability phase II (ArCS-II; grant no.
JP- MXD1420318865) project. DB was supported by FORMAS (Grant No. 2018-01794), the European Union
(H2020 Grant No. 101015825; TRIAGE) and the Swedish Research Council VR (Grant No. 2022-03841). PR
acknowledges funding from the European Union's Horizon 2020 research and innovation program under Grant
Agreement No. 101003536 (ESM2025-Earth System Models for the Future) and from the FRS-FRNS PDR
project T.0191.23 CH4-lakes. Annual, gap-filled and harmonized NGHGI uncertainty estimates for the EU were
provided by the EU GHG inventory team (European Environment Agency and its European Topic Centre on
Climate change mitigation). RL acknowledges funding from French state aid, managed by ANR under the
"Investissements d'avenir" programme (ANR-16-CONV-0003). WZ was supported by grants from the Swedish
Research Council VR (2020-05338) and Swedish National Space Agency (209/19).

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
