# Peer review of "Comparison of observation- and inventory- based methane emissions for eight large global"

_Earth System Science Data, 2023_

## Author Comment (AC1)

Reconciliation of observation- and inventory- based methane emissions for eight large global emitters

Referee #1

*Petrescu et al. compiled observation- and inventory-based methane emission estimates for 8 large global emitters. These data from different sources all come with inconsistent formats and sector partitions. The authors harmonized the dataset with a consistent sector partition, so different estimates can be compared properly, which is valuable. However, I have several concerned that (1) the paper contains substantial discussions that are not directly related to the dataset, which distracts the main purpose;*

We are grateful to Reviewer#1 for taking time to read and comment on this draft and we are pleased to hear that, even if currently does not recommend the paper for its publication, he finds the work valuable.

Regarding the first concern (1): The paper was built on the results from the three-year EU funded CoCO2 project. Section "4. Discussion and recommendations on reconciliation procedures" shares our experiences of performing reconciliation and how this can be improved, which comes from a project deliverable on a blueprint for a decision support system to be used in an eventual CO2MVS. We believe this adds value as it builds on dialogues with a much broader community of users, e.g. scientists, inventory agencies, policy makers etc. taking into account their opinions and needs when it comes to comparisons between the two approaches.

Having said that, we will try to shorten and make the discussion more concise. If the reviewers and the editor in charge will agree, we would still like to include the section 4 discussion as an important input on user engagement guidelines to be produced in the future under different communities (WMO and IG3IS, and EU funded projects).

*(2) the statement that observation- and inventory-based methane emissions are reconciled is an exaggeration and is misleading, so, I cannot recommend the paper, in its current form, to publish in ESSD. Below explains my main comments.*

Regarding Reviewer's concern on the use of the word "Reconciliation" and that it may be an "exaggeration and misleading". This likely depends on the context in which the word is used. In our case we see the word as what we are doing (reconciliation) as opposed to what was successfully done (reconciled): reconciliation is the process or action of making one dataset comparable with another to assess their consistency. Reconciliation is very much the process, not the outcome.

In this respect, we tried to obtain consistent results from both BU and TD estimates, and the compatibility between BU and TD products implies harmonization of the results, concepts and definitions. As we are comparing BU and TD approaches, I believe we can, in terms of our purpose, and including the available data we have, present it as an attempt to reconcile the two approaches.

We will add a clearer definition of what we mean by reconciliation in the revision.

**Main comments**

*The objective of the paper is not clearly stated. As a data paper, one would think that main objective is to describe the dataset. In the case of this paper, since the data are taken from other studies, the key is to describe how diverse data are harmonized and what the harmonized data tell us.*

Thank you for your observation. It is true that one should explain the harmonizing procedure and for that we show how diverse the data is (Table 2 and Figure 5) and for comparability we also try our best to harmonize it (Table 3 and Figure 6). We will rewrite this section to better explain it.

*However, Section 4 contains substantial discussion that is very general and not directly related to the dataset, which is distractive.*

In our opinion, this section is really important and constitutes a message from a user engagement perspective. Ideally, we want close collaboration between the scientific community and the inventory community, if their emissions are to be verified. For EU Member States to be able to use scientific products to complement their inventories, monitoring and verification procedures are highly important. Key issues in reconciliation identified during our research (CoCO2) are particularly relevant when inventory agencies want to use our scientific data and analysis to complement their inventory work.

*The recommendations given in the end are random and not backed up by the findings of the paper. I would suggest that the authors clearly state their objective and organize the content around that.*

We thank the reviewer for this suggestion. These recommendations are not random, are the summary of Section 4 and the findings from long comparisons exercises between reported data and scientific BU and TD estimates. We agree to the reviewer's suggestion and will better link the findings to the objectives.

*Specifically, in Line 134-135, the authors claimed that the paper "aims to inform and attract attention of the use of the results for diverse climate stakeholder needs beyond research", which is a very good statement of the objective. However, it was then not discussed anywhere in the method and results. It is not never explicitly explained what prevented stakeholders from using the existing methane emission data, why this dataset compiled by the authors would be more appealing, and what efforts have been made to make the data easy to use. Moreover, I checked out the data in the repository. The datasheets still look very complicated to me, and I am not sure that non-researchers can easily find the information they needed. Anyway, more explanations are needed if the above statement is indeed the objective of the paper.*

Thank you for your positive comment. Indeed the objective of this paper is linked to user engagement. Therefore, we will make sure we will incorporate adequate discussion regarding the involvement of stakeholders in this process. Regarding the format and complicated data from the repository, according to the policy of ESSD, we provide the numbers behind the figures and, as previously done for other publications, we provide the readers with the timeseries of each dataset per country. We believe it ensures replicability for anyone who can use excel/python in a basic/intermediate way.

*2. "Reconciliation of observation- and inventory-based methane emissions" is used as the title and presented as the main finding of the paper. I find it an overstatement. The paper presented "total inversion methane emissions > BU anthropogenic emissions" as a discrepancy, which was reconciled by considering "posterior total flux from inversions (roughly) = BU anthropogenic flux + BU natural flux" at the national scale. However, this level of consistency/reconciliation is not surprising at all. Why would anyone want to directly compare total methane emissions from an inversion with just anthropogenic emissions from an inventory and ignore natural sources? So, I feel that the title exaggerated what was found in the paper. It is very likely that a more in-depth investigation into the data would identify significant discrepancies between the observation- and inventory- based methane emissions.*

As already explained above, the title includes the term reconciliation from the perspective of the process of comparing two or more sets of results, ensuring their accuracy and consistency. For

consistency, we performed and explained the harmonization procedures, regarding partitions and priors present in each dataset (Tables 2 and 3).

We also discuss not only total TD versus anthropogenic BU, but we include the natural emissions as well. There is a section dedicated to the harmonization of natural emissions from TD partitions (Table 4). In this respect, the title talks about BU and TD reconciliation, and takes into account all available CH4 sources. This comparison, which we believe is complete, identifies already inconsistencies which are summarized in the section 3.4.2. If the scientific community wants to assist Members States in improving reporting (e.g. improved EFs, detect and quantify accurately the hot-spot emissions, spatial distribution of emissions), and supporting the national reporting agencies in using atmospheric data to complement and complete their NGHGIs, we should be able to explain differences and take into account all existing sources. There will be always discrepancies between datasets, as they do not include the same partitions, however if we are able to emphasize and quantify them, this will clarify the aim of this work. We agree that this harmonization section is not very well structured, so we will address again these issues and give a clearer and more organized explanation.

*A more important questions is the reliability of the comparison made here between the inversions and inventories. Inversions are regarded as independent top-down verification of bottom-up inventories. But they are not. All of these inversions rely on prior information and therefore not independent of bottom-up emission inventories.*

The reviewer is right that inversions do not provide GHG emission estimates that are independent from the prior that is used. Because different inventories rely on different methodologies and Tiers (e.g., IPCC defaults (Tier 1) versus country-specific data (Tiers 2 and 3)), priors used in inversions are often different from values reported in NGHGIs, which already provides some independence. Furthermore, and more importantly, the atmospheric measurements do provide independent information about emissions. Therefore, the evaluation against atmospheric measurements and adjustment of emissions needed to bring the atmospheric transport model in agreement with them is based on independent information. The inversion gives an indication of how far the atmospheric data suggests to move away from the prior.

Having said that, we believe it is useful to compare the two approaches, as long as we highlight these dependences and explain the differences.

*Comparing bottom-up inventories and inversions without characterizing this dependence makes it difficult to judge whether the two are actually consistent. For example, we do not know whether the agreement of the EU emission trend from various inversions was due to strong observation evidence, or due to similar prior information used by the inversions and a weak observation constraint. We also do not know if the disagreement in the USA emission trend was due to a weak observation constraint and different prior information. If this is the case, it makes little sense to talk about the consistency between the inversions and bottom-up inventories in terms of the USA emission trend.*

As previously discussed in Petrescu et al., 2020 (Figure 4) regarding trends for the EU27 from BU estimates, we agree that EU trends have similarities due to prior information used by inverse systems, which are coming from the BU results.

Both BU inventories and the NGHGI use similar activity data and, to varying extents, the default EFs reported in the IPCC (2006) guidelines, meaning that the estimates are predestinated to agree rather well. Thus, the spread in all BU estimates may not be indicative of the uncertainty.

The few inconsistencies between CH4 BU estimates and NGHGI are mainly caused by different methodologies in calculating emissions as highlighted in Petrescu et al. (2020, 2021).

For the USA, we explained why the trends for the CTE and MIROC are different from those from CAMS and TM5-4DVAR (Tropomi based). This is explained in the discussion after Figure 4 (as well as in the Supplementary Information) and this is due to the fact that the priors for oil & gas emissions for CTE and MIROC are based on the GAINS model (which shows a similar trend in Figure 3) while the other models use EDGARv6.

For a good overview of differences between priors, we have in the Zenodo repository an excel file which includes a spreadsheet named "Priors" summarizing each model and the priors they use for different partitions. https://doi.org/10.5281/zenodo.10276087. We believe that in most countries, these are the reasons for differences.

**Minor comments:**

We thank the reviewer for these comments. Some of the comments were also highlighted by the Reviewer in the major comments section above, and we believe we answered accordingly.

The minor issues we will address accordingly in the revised version of the paper.

*Line 128-129: The statement indicates that achieving the climate goal will automatically lead to gains in areas of energy, food, etc., which can be misleading. In fact, controlling methane emissions may pose significant challenges to energy and food security. I suggest rephasing the statement to be more balanced.*

*Line 190: Spell out LULUCF at the first appearance.*

*Line 191: Missing information. "...which according to the ? are defined as... "*

*Line 199: Period sign before "Furthermore".*

*Line 241: What is IPPU? Spell it out and explain if necessary.*

*Line 291: e.g. -> i.e.*

*Line 301-302: It may be useful to report the rate of reduction in USA emissions, as a comparison to the EU value reported above.*

*Line 339: Perhaps be more specific that Russian CH4 emissions remained rather low "relative to its pre-2000 levels"? Compared to other countries, Russian emissions are not low at all.*

*Line 383: What is AD? Activity data? Spell out and explain if necessary.*

*Line 464-473: The paragraph appears to be out of context. All the remainder of the section discusses the BU and TD comparison (in terms of both average emissions and trends), while this paragraph talks generally about sectors driving CH4 growth.*

*Line 484-490: This result shows that the emission trends derived from the inversions are strongly dependent on the prior choice, indicating that the atmospheric observations used in these inversions are inadquate to constrain the emission trend. The relatively good agreement of emission trends in other countries (e.g., EU) also does not provide strong evidence, because the agreement can be driven by similar prior information.*

*Line 501-502: Again, this may be due to different choices of prior emissions.*

*Figure 6 and Line 600: Biomass burning is considered anthropogenic in Figure 6 but natural in Line 600. I understand both anthropogenic and natural processes contribute to biomass burning. However, the current description is unclear and confusing. A clear description and terminology should be given to distinguish its anthropogenic and natural components.*

*Table 3 and 4: The orders of inversions are listed differently in the two tables, making it difficult to compare.*

*Table 4: For an inversion, there is a difference between "missing" and "unreported" sources. If these natural sources are included in the model simulation but are not reported as results (for example because they are not optimized by the algorithm), it makes sense to use BU estimate in place in order to compare "apple to apple". However, if these sources are not included in the prior simulation, the posterior total flux inferred from observations may still implicitly include their contributions because the observation sees the total flux, although the fluxes from these sources can be mis-attributed to other sources. If this is the case, adding BU estimates to inversion estimates will actually lead to inconsistent comparison. Therefore, it is important to distinguish between "missing" or "unreported" sources, or discuss the complication.*

*Line 701-707: The discussion on city-level and even event- or facility-level inversion is irrelevent to this study. The entire paper is on national emissions. It is still unclear how information is integrated on these very different scales.*

*Line 723-724: Worden et al. (2023) provides a framework to properly compare inventory and observation-based inversions.*

*Worden, J. R., Pandey, S., Zhang, Y., Cusworth, D. H., Qu, Z., Bloom, A. A., et al. (2023). Verifying methane inventories and trends with atmospheric methane data. AGU Advances, 4, e2023AV000871.*

*Line 724: "Some" attempts*

*Line 770: I think this is a very good recommendation. However, it is inadequately discussed and justified in the paper. A reader may want to learn the justification of this and other recommendations.*

We will add to each of these recommendations links to the discussion and additional explanations.

*Line 738-740, Line 776: What do you mean by measurement of fluxes? Please be explicit whether you talking about regional fluxes or fluxes of specific sources? In the case of latter, it is actually emission factors rather than fluxes that are directly useful for a better prior estimate. Moreover, activity data, in many cases, are also bottle-necks for better priors and better uncertainty estimates, in addition to emission factors.*

*Line 922-930: Redundant reference information.*

*Line 832: Inversions are not entirely independent data, as they rely on the prior information.*

---

## Author Comment (AC2)

Reconciliation of observation- and inventory- based methane emissions for eight large global emitters

Referee #2

*This study analyzes methane emissions and their annual variability using multiple bottom-up (BU) inventories and top-down (TD) studies for the European Union (EU) and seven additional countries with substantial emissions. The authors specifically aim to reconcile the BU and TD results by harmonizing the source sectors, and also offer insights to enhance the intercomparison between BU and TD studies. Acknowledging the significance and substantial workload involved in synthesizing a large volume of data, I believe there is considerable room to enhance the discussion, clarity, and readability of the study. Here are some suggestions*

We thank Reviewer #2 for valuable suggestions and we acknowledge that the paper lacks some detailed explanations. We will improve the discussion and provide better clarity to the findings.

*1.The purpose for such an "update" needs to be clearer. The authors state that "this study updates earlier syntheses (Petrescu et al., 2020, 2021, 2023) and provides a consolidated synthesis of CH4 emissions". What's the need for the update? Compared to the previous syntheses, what new focus, data, and methods have been incorporated in this update?*

Previous syntheses were focused on EU28, we have updated that and expanded to major global emitters, as a first step towards building a more global CO2MVS capacity and to independently assess the progress of countries towards their climate targets. However, we agree that this can be made clearer in the introduction of the paper as well as stating the objective.

*2. The article does not describe the scope of the inversion results included in the discussion. Why were only the inversion results in Table 1 included in the discussion? What were the reasons for selecting these inversion results?*

We focused on systems already developed and/or included in previous projects (e.g., VERIFY) and further developed under the CoCO2 project. During the writing of this paper, we also presented published results in lieu of more recent inversions (Worden et al., 2019 and Nesser et al., 2023). We can explain this more clearly in the revision.

*Existing literature contains far more TD inversion studies than those discussed in the paper, with detailed discussions of the regions of interest and providing inversion results for interannual variations.*

Our goal was not to include everything what's available out there, and this choice was based on previous collaborations. We believe that the TD inversions selected for this study are also the most representative for the selected domain. We also wanted to avoid duplication of work done by other colleagues (Saunois et al., will soon submit the new updated CH4 global budget).

*These existing studies highlight inconsistency between BU and TD studies for many hotspot regions, such as oil and gas methane emissions in North America, coal emissions in East Asia, and wetland emissions in Europe, North America, and Africa, yet these are not reflected in the paper's discussion. This is a significant weakness. A large portion of these existing inversion studies can be found in Jacob (2022)'s review paper.*

Thank you for your suggestion. We also highlight inconsistencies and reference some of these studies. We will adapt the discussion and include the references accordingly.

*3. Figure 5 and Figure 6: why the uncertainty of CTE-GCP is so large? The uncertainty of total emissions cannot be estimated as the sum of uncertainty from each sector.*

The CTE uncertainty does not represent the sum of the uncertainties for each sector. The reason why they are so large can be found in the inversion and its setup, and in the way CTE calculates the uncertainties. They use an Ensemble Kalman Filter with 500 ensemble members in the optimisation, which means that they create 500 different "versions" of the prior emissions using the uncertainties assigned to them (perturbing the prior emissions by taking a random sample). The prior uncertainty is then the standard deviation of these 500 members in a given domain. The optimised result is defined as the average of the 500 members (after optimisation) and the uncertainty is its standard deviation.

The uncertainties can be quite large, depending on how one choses the prior uncertainties and how many observations there were to constrain the optimised emissions. China is a rather large region with few observation sites, similar to Brazil, Indonesia and DR Congo (those countries where CTE shows really high uncertainties in Figure 5 and 6) but if one compares the prior uncertainties with the posterior uncertainties, the optimized uncertainties have decreased.

We would indeed get more information by comparing the posterior uncertainty to the prior uncertainty. However, it still reflects the knowledge and assumptions made for the domain under study, which is an advantage of using the Ensemble Kalman Filter. Here, we show a map of observation sites constraining the inversions (we can also add this explanation to the SI, including the figure).

[Figure]

Fig. Region masks used to calculate the emissions and uncertainties and insitu stations used in the inversion (yellow dots)

*4. Figure 7: It is very difficult to understand the bar of "BU Natural" and "TD Natural"? Why were the same BU anthropogenic emissions subtracted from the total emissions? Please enhance the clarity of this figure for better readability.*

The BU natural and TD natural represent the respective natural emissions from BU and TD products. As explained in Table 4, not all TD products include/report the same partitions for the natural emissions. For harmonization purpose, we added BU estimates to those TD datasets which do not report this information, as hashed bar on top, which represent the missing natural partition.

We do not subtract any BU anthropogenic from the total, we just present the reported sources: anthropogenic, natural and total.

*5. It would be great to have a figure that highlights the regions and emission sectors where the differences between current top-down and bottom-up results are most significant.*

Thank you for this suggestion. The problem is that we do not have a really complete dataset to be able to make such a figure. There are a lot of gaps in the UNFCCC reporting for the non-Annex I countries. We believe that Figure 7 shows rather good agreement.

We could provide a table with the difference of the median of the BU+Natural and TD, which will give an idea of how big the differences are. This table could contain the countries in the rows, then the columns for median and range of BU, median and range of Natural, BU+Natural (median and range based on sum of squares error), TD median and range, and a final column on the difference. We could then discuss if the TD lies outside of the BU+Natural range, and could also see if they are within uncertainties, and reconciled.

*6. There are too many abbreviations in the text. It is recommended to include a table that summarizes the corresponding full forms of these abbreviations, maybe in the Appendix.*

Thank you for your observation, we will provide such a table, however we did not include an Appendix to the paper, we will add it to the Supplementary Information file.

*7. Line 199 : missing a ".".*

*Reference*

*Jacob, D.J., D.J. Varon, D.H. Cusworth, P.E. Dennison, C. Frankenberg, R. Gautam, L. Guanter, J. Kelley, J. McKeever, L.E. Ott, B. Poulter, Z. Qu, A.K. Thorpe, J.R. Worden, and R.M. Duren, Quantifying methane emissions from the global scale down to point sources using satellite observations of atmospheric methane, Atmos. Chem. Phys., 22, 9617–9646, https://doi.org/10.5194/acp-22-9617-2022, 2022.*

Thank you, we will update the references accordingly.

---

## Author Response (AR1)

"Reconciliation of observation- and inventory- based methane emissions for eight large global emitters" by AMR Petrescu et al., 2024

The answers and new line numbers refer to the Track Change document.

**Referee #1**

*Petrescu et al. compiled observation- and inventory-based methane emission estimates for 8 large global emitters. These data from different sources all come with inconsistent formats and sector partitions. The authors harmonized the dataset with a consistent sector partition, so different estimates can be compared properly, which is valuable. However, I have several concerned that (1) the paper contains substantial discussions that are not directly related to the dataset, which distracts the main purpose;*

We are grateful to Reviewer#1 for taking time to read and comment on this draft and we are pleased to hear that, he finds the work valuable.

Regarding the first concern (1): The paper was built on the results from the three-year EU funded CoCO2 project. Section "4. Discussion and recommendations on reconciliation procedures" shares our experiences of performing reconciliation and how this can be improved, which comes from a project deliverable on a blueprint for a decision support system to be used in an eventual CO2MVS. We believe this adds value as it builds on dialogue with a much broader community of users, e.g. scientists, inventory agencies, policy makers etc. taking into account their opinions and needs when it comes to comparisons between the two approaches. Nevertheless, we have substantially reduced this section to keep the paper more focused as suggested by the reviewer.

We added to the abstract (L49-50) a sentence on the aim of the study and at the end of the introduction (L158-172) a new paragraph explaining the link to the work done in the CoCO2 project (user engagement). Section 4 is greatly reduced, but elements remain due to importance to related studies and user involvement (e.g., WMO and IG3IS, and EU funded projects).

*(2) the statement that observation- and inventory-based methane emissions are reconciled is an exaggeration and is misleading, so, I cannot recommend the paper, in its current form, to publish in ESSD. Below explains my main comments.*

Regarding Reviewer's concern (2) on the use of the word "Reconciliation" and that it may be an "exaggeration and misleading". This likely depends on the context in which the word is used. In our case we see the word as what we are doing (reconciliation) as opposed to what was successfully done (reconciled): reconciliation is the process or action of making one dataset comparable with another to assess their consistency. Reconciliation is very much the process, not the outcome.
Nevertheless, given the ambiguity in the term reconciliation, and that it was used relatively few times throughout the article, we have decided to use the word "comparison" instead. This means the title changes slightly, and a few mentions of the word reconciliation in the article are removed.

The start of section 2 explains the context a little better and we added the following paragraph: *"In this work we focus on comparing BU and TD emission estimates. The 'reconciliation process' described in this work is the action of making one dataset comparable with another to assess their consistency. In this respect, we attempt to obtain consistent results from both BU and TD estimates, through harmonization of the results, concepts and definitions. After the reconciliation process, the estimates do not necessarily agree, representing uncertainties in the different methods and datasets. We now describe the key data and methods used in our analysis".*

**Main comments**

*The objective of the paper is not clearly stated. As a data paper, one would think that main objective is to describe the dataset.*

Regarding the clearness of the objectives, we detailed this information in the Abstract L48-50 and new lines L69-70

*In the case of this paper, since the data are taken from other studies, the key is to describe how diverse data are harmonized and what the harmonized data tell us.*

Thank you for your observation. It is true that one should explain the harmonizing procedure and for that we show how diverse the data is (Table 2 and Figure 5) and for comparability we also try our best to harmonize it (Table 3 and Figure 6). We have tried to improve the wording throughout the article to help clarify these issues and challenges.

*However, Section 4 contains substantial discussion that is very general and not directly related to the dataset, which is distractive. The recommendations given in the end are random and not backed up by the findings of the paper. I would suggest that the authors clearly state their objective and organize the content around that.*

In our opinion, this section is really important and constitutes a message from a user engagement perspective. Ideally, we want close collaboration between the scientific community and the inventory community, if their emissions are to be verified. For EU Member States to be able to use scientific products to complement their inventories, monitoring and verification procedures are highly important. Key issues in reconciliation identified during our research (CoCO2) are particularly relevant when inventory agencies want to use our scientific data and analysis to complement their inventory work.

Nevertheless, we have greatly shortened this section, and kept focus on issues that are discussed in the article (priors, uncertainty, natural emissions, etc.). We believe this gives valuable lessons for future studies in this area.

*Specifically, in Line 134-135, the authors claimed that the paper "aims to inform and attract attention of the use of the results for diverse climate stakeholder needs beyond research", which is a very good statement of the objective. However, it was then not discussed anywhere in the method and results. It is not never explicitly explained what prevented stakeholders from using the existing methane emission data, why this dataset compiled by the authors would be more appealing, and what efforts have been made to make the data easy to use. Moreover, I checked out the data in the repository. The datasheets still look very complicated to me, and I am not sure that non-researchers can easily find the information they needed. Anyway, more explanations are needed if the above statement is indeed the objective of the paper.*

Thank you for your positive comment. Indeed the objective of this paper is linked to user engagement. We have added text in various places that links to the new reporting rules for non-Annex I countries and the need to build capacity. Yes, the data is complex, but these are complex issues. A part of the motivation of the paper, and indeed Section 4, is to highlight these complexities and how they can potentially be alleviated. These data, whilst complex, may lead to the greatest marginal gains for non-Annex I countries.

Regarding the format and complicated data from the repository, according to the policy of ESSD, we provide the numbers behind the figures (https://doi.org/10.5281/zenodo.12582667) and, as previously done for other publications, we provide the readers with the timeseries of each dataset per country. We believe it ensures replicability for anyone who can use excel/python in a basic/intermediate way.

*2. "Reconciliation of observation- and inventory-based methane emissions" is used as the title and presented as the main finding of the paper. I find it an overstatement. The paper presented "total inversion methane emissions > BU anthropogenic emissions" as a discrepancy, which was reconciled by considering "posterior total flux from inversions (roughly) = BU anthropogenic flux + BU natural flux" at the national scale. However, this level of consistency/reconciliation is not surprising at all. Why would anyone want to directly compare total methane emissions from an inversion with just anthropogenic emissions from an inventory and ignore natural sources? So, I feel that the title exaggerated what was found in the paper. It is very likely that a more in-depth investigation into the data would identify significant discrepancies between the observation- and inventory-based methane emissions.*

As explained above, we have used "comparison" in the title to avoid misinterpretation of reconciliation.

We agree with the reviewer here, we do not want to do apples versus oranges comparisons. For consistency, we performed and explained the harmonization procedures, regarding partitions and priors present in each dataset (Tables 2 and 3). We have now done the comparisons between BU and TD after removing natural emissions, to make the comparison clearer. There is a section dedicated to the harmonization of natural emissions from TD

partitions (Table 4). This comparison, which we believe is complete, identifies already inconsistencies which are summarized in the section 3.5.

If the scientific community wants to assist Members States in improving reporting (e.g. improved EFs, detect and quantify accurately the hot-spot emissions, spatial distribution of emissions), and supporting the national reporting agencies in using atmospheric data to complement and complete their NGHGIs, we should be able to explain differences and take into account all existing sources. There will be always discrepancies between datasets, as they do not include the same partitions, however if we are able to emphasize and quantify them, this will clarify the aim of this work.

*A more important questions is the reliability of the comparison made here between the inversions and inventories. Inversions are regarded as independent top-down verification of bottom-up inventories. But they are not. All of these inversions rely on prior information and therefore not independent of bottom-up emission inventories.*

The reviewer is right that inversions do not provide GHG emission estimates that are completely independent. The priors used in inversions are different from values reported in NGHGIs. Most inversions have used EDGAR as a prior emission dataset. As shown in the BU comparisons, EDGAR does not always agree with UNFCCC. The priors also need to be gridded, which means that UNFCCC NGHGIs cannot be used directly in the inversions (EDGAR is gridded).

Furthermore, and more importantly, the atmospheric measurements do provide independent information about emissions. The atmospheric measurements provide an additional constraint, which when binding, will lead to posterior estimates more in line with reality. It is of most interest how the prior shifts to the posterior, and how these compare to the NGHGIs. The uncertainty information provides additional information on how important the atmospheric observations were in shifting the prior to posterior. Uncertainty information is something very hard to extract from the inversion modellers, and we discuss this in Section 4.

Having said that, we believe it is useful to compare the two approaches, highlighting these dependencies and explaining the cause of the differences. Whilst perhaps not evident in the text, when comparing BU and TD, these sorts of issues are constantly on our mind. In fact, section 4 was largely about dealing with the issues raised by the reviewer.

*Comparing bottom-up inventories and inversions without characterizing this dependence makes it difficult to judge whether the two are actually consistent. For example, we do not know whether the agreement of the EU emission trend from various inversions was due to strong observation evidence, or due to similar prior information used by the inversions and a weak observation constraint. We also do not know if the disagreement in the USA emission trend was due to a weak observation constraint and different prior information. If this is the case, it makes little sense to talk about the consistency between the inversions and bottom-up inventories in terms of the USA emission trend.*

Broadly speaking, the reviewer is correct and summarises the challenges with inversions. Yes, we need to know the priors, the uncertainties on the prior and posterior, and how they have been changed by the atmospheric data. This gives us confidence in the results. A part of the motivation for section 4 was to discuss these challenges. We build confidence when multiple lines of evidence align (e.g., BU, multiple TD evidence based on different priors, etc). One could argue we have greater confidence in the EU given multiple lines of evidence coincide, while this is not the case in the US.

As previously discussed in Petrescu et al., 2020 (Figure 4) regarding trends for the EU27 from BU estimates, we agree that EU trends have similarities due to prior information used by inverse systems, which are coming from the BU results. Both BU inventories and the NGHGI use similar activity data and, to varying extents, the default EFs reported in the IPCC (2006) guidelines, meaning that the estimates are predestinated to agree rather well. Thus, the spread in all BU estimates may not be indicative of the uncertainty. The few inconsistencies between CH4 BU estimates and NGHGI are mainly caused by different methodologies in calculating emissions as highlighted in Petrescu et al. (2020, 2021).

For the USA, we explained why the trends for the CTE and MIROC are different from those from CAMS and TM5-4DVAR (Tropomi based). This is explained in the discussion after Figure 4 (as well as in the Supplementary Information) and this is due to the fact that the priors for oil & gas emissions for CTE and MIROC are based on the GAINS model (which shows a similar trend in Figure 3) while the other models use EDGARv6.

For a good overview of differences between priors, we have in the Zenodo repository an excel file (which has been updated to v2) which includes a spreadsheet named "Priors" summarizing each model and the priors they use for different partitions. We believe that in most countries, these are the reasons for differences.

Data behind figures are available here: https://doi.org/10.5281/zenodo.12582667.

**Minor comments:**

We thank the reviewer for these comments. Some of the comments were also highlighted by the Reviewer in the major comments section above, and we believe we answered accordingly.

The minor issues we will address accordingly in the revised version of the paper.

*Line 128-129: The statement indicates that achieving the climate goal will automatically lead to gains in areas of energy, food, etc., which can be misleading. In fact, controlling methane emissions may pose significant challenges to energy and food security. I suggest rephasing the statement to be more balanced.*

L136-140: we changed the sentence as following: "Achieving this goal will drive significant gains, through specific energy and agriculture defined pathways including innovative actions, national targeted policies, and green climate funds to help smallholder farmers (https://www.state.gov/global-methane-pledge-from-moment-to-momentum/)."

*Line 190: Spell out LULUCF at the first appearance.*

Now L223 done

*Line 191: Missing information. "...which according to the ? are defined as... "*

Now L 225 Added IPCC guidelines

*Line 199: Period sign before "Furthermore".* Corrected

*Line 241: What is IPPU? Spell it out and explain if necessary.*

Now L278 explanation added

*Line 301-302: It may be useful to report the rate of reduction in USA emissions, as a comparison to the EU value reported above.* Now L338, added the word rate

*Line 339: Perhaps be more specific that Russian CH4 emissions remained rather low "relative to its pre-2000 levels"? Compared to other countries, Russian emissions are not low at all.* thank you, we replaced as suggested (now L385)

*Line 383: What is AD? Activity data? Spell out and explain if necessary.* We added the explanation throughout

*Line 464-473: The paragraph appears to be out of context. All the remainder of the section discusses the BU and TD comparison (in terms of both average emissions and trends), while this paragraph talks generally about sectors driving CH4 growth.*

The reviewer is right, we move this paragraph in the introduction where we mention the CH4 growth and drivers noticed in the last years (now L111-120)

*Line 484-490: This result shows that the emission trends derived from the inversions are strongly dependent on the prior choice, indicating that the atmospheric observations used in these inversions are inadquate to constrain the emission trend. The relatively good agreement of emission trends in other countries (e.g., EU) also does not provide strong evidence, because the agreement can be driven by similar prior information.*

This would depend on the uncertainty reduction in the inversion. For example, if the prior had 20% uncertainty, and the posterior had 5% uncertainty, then this would indicate the observations have improved the estimate. The challenge is that it is very difficult to get the prior and posterior uncertainties from each modelling group. This issue is discussed in Section 4.

*Line 501-502: Again, this may be due to different choices of prior emissions.*

Yes, we agree. Please see previous comment.

*Figure 6 and Line 600: Biomass burning is considered anthropogenic in Figure 6 but natural in Line 600. I understand both anthropogenic and natural processes contribute to biomass burning. However, the current description is unclear and confusing. A clear description and terminology should be given **to distinguish its anthropogenic and natural components.***

Yes, this is the challenge we have with harmonization, as each inversion does this differently. We do not consider the biomass burning anthropogenic, we are constrained to add it to the anthropogenic emissions together with rice because some inversions do not report it separately.Therefore, we chose for comparison purposes, to add it to the anthropogenic+rice+BB group.

*Table 3 and 4: The orders of inversions are listed differently in the two tables, making it difficult to compare.*

We rearranged the order of inversions in Tables 2, 3 and 4 for a better visual comparison.

*Table 4: For an inversion, there is a difference between "missing" and "unreported" sources. If these natural sources are included in the model simulation but are not reported as results (for example because they are not optimized by the algorithm), it makes sense to use BU estimate in place in order to compare "apple to apple". However, if these sources are not included in the prior simulation, the posterior total flux inferred from observations may still implicitly include their contributions because the observation sees the total flux, although the fluxes from these sources can be mis-attributed to other sources. If this is the case, adding BU estimates to inversion estimates will actually lead to inconsistent comparison. Therefore, it is important to distinguish between "missing" or "unreported" sources, or discuss the complication.*

Thank you for highlighting this issue. It is true that some do not report it separate and some do not even include it in their priors. Therefore, according Zenodo priors table we corrected some of the double counted emissions (e.g for CTE-GCP2021 we removed the BB and geological emissions as considered in priors, and for CAMS and TM5-4DVAR (both runs) we removed the termites, oceans and soils sink emissions; for the latter we added the geological emissions. See new table 4.

*Line 701-707: The discussion on city-level and even event- or facility-level inversion is irrelevent to this study. The entire paper is on national emissions. It is still unclear how information is integrated on these very different scales.*

This section is now shorter.

*Line 723-724: Worden et al. (2023) provides a framework to properly compare inventory and observation-based inversions.*

*Worden, J. R., Pandey, S., Zhang, Y., Cusworth, D. H., Qu, Z., Bloom, A. A., et al. (2023). Verifying methane inventories and trends with atmospheric methane data. AGU Advances, 4, e2023AV000871.*

Thank you for your suggestion. We do agree with the proposed methodology of Worden et al., however our intention was not to replicate it but to present the CoCO2 project results and the available data and provide explanations on observed deviations from the NGHGIs and BU vs TD.

*Line 724: "Some" attempts*

Thank you, we corrected.

*Line 770: I think this is a very good recommendation. However, it is inadequately discussed and justified in the paper. A reader may want to learn the justification of this and other recommendations.*

Thank you. Section 4 was shortened, and the list of recommendations was removed, but were kept in the overall section 4.

*Line 738-740: What do you mean by measurement of fluxes? Please be explicit whether you talking about regional fluxes or fluxes of specific sources? In the case of latter, it is actually emission factors rather than fluxes that are directly useful for a better prior estimate. Moreover, activity data, in many cases, are also bottle-necks for better priors and better uncertainty estimates, in addition to emission factors.*

We refer here to in-situ data, observation-based fluxes from networks like FLUXNET, ICOS towers and mobile platforms/aircraft or chambers (now L776)

*Line 922-930: Redundant reference information.*

We deleted the double reference.

*Line 832: Inversions are not entirely independent data, as they rely on the prior information.*

This is true, and we provided the explanation above, under issue 2.

**Referee #2**

*This study analyzes methane emissions and their annual variability using multiple bottom-up (BU) inventories and top-down (TD) studies for the European Union (EU) and seven additional countries with substantial emissions. The authors specifically aim to reconcile the BU and TD results by harmonizing the source sectors, and also offer insights to enhance the intercomparison between BU and TD studies. Acknowledging the significance and substantial workload involved in synthesizing a large volume of data, I believe there is considerable room to enhance the discussion, clarity, and readability of the study. Here are some suggestions*

We thank Reviewer #2 for valuable suggestions and we acknowledge that the paper lacks some detailed explanations. We will improve the discussion and provide better clarity to the findings.

*1.The purpose for such an "update" needs to be clearer. The authors state that "this study updates earlier syntheses (Petrescu et al., 2020, 2021, 2023) and provides a consolidated synthesis of CH4 emissions". What's the need for the update? Compared to the previous syntheses, what new focus, data, and methods have been incorporated in this update?*

Previous syntheses were focused on EU28, we have updated that and expanded to major global emitters, as a first step towards building a more global CO2MVS capacity and to independently assess the progress of countries towards their climate targets. However, we agree that this can be made clearer in the introduction of the paper as well as stating the objective.

We highlight now that we expand the study to consider the EU and seven additional countries. We also added in the abstract a clearer statement of the aim and at the end of the introduction explained that this builds on the work done in the CoCO2 project (user engagement) (L158-173)

*2. The article does not describe the scope of the inversion results included in the discussion. Why were only the inversion results in Table 1 included in the discussion? What were the reasons for selecting these inversion results?*

We focused on systems already developed and/or included in previous projects (e.g., VERIFY) and further developed under the CoCO2 project. During the writing of this paper, we also presented published results in lieu of more recent inversions (Worden et al., 2019 and Nesser et al., 2023). We can explain this more clearly in the revision. We added L158-L173 at the end of the Introduction stating the CoCO2 result-based approach.

*Existing literature contains far more TD inversion studies than those discussed in the paper, with detailed discussions of the regions of interest and providing inversion results for interannual variations.*

Our goal was not to include everything what's available out there, and this choice was based on previous collaborations. We believe that the TD inversions selected for this study are also the most representative for the selected domain. We also wanted to avoid duplication of work done by other colleagues (Saunois et al., will soon submit the new updated CH4 global budget).

*These existing studies highlight inconsistency between BU and TD studies for many hotspot regions, such as oil and gas methane emissions in North America, coal emissions in East Asia, and wetland emissions in Europe, North America, and Africa, yet these are not reflected in the paper's discussion. This is a significant weakness. A large portion of these existing inversion studies can be found in Jacob (2022)'s review paper.*

Thank you for your suggestion. We also highlight inconsistencies and reference some of these studies in section 4 discussions. We adapted the discussion and included the references accordingly, also in the Introduction.

*3. Figure 5 and Figure 6: why the uncertainty of CTE-GCP is so large? The uncertainty of total emissions cannot be estimated as the sum of uncertainty from each sector.*

Please see below the explanation and Fig. Region masks used to calculate the emissions and uncertainties and in-situ stations used in the inversion (yellow dots).
The CTE uncertainty does not represent the sum of the uncertainties for each sector. The reason why they are so

large can be found in the inversion and its setup, and in the way CTE calculates the uncertainties. They use an Ensemble Kalman Filter with 500 ensemble members in the optimisation, which means that they create 500 different "versions" of the prior emissions using the uncertainties assigned to them (perturbing the prior emissions by taking a random sample). The prior uncertainty is then the standard deviation of these 500 members in a given domain. The optimised result is defined as the average of the 500 members (after optimisation) and the uncertainty is its standard deviation. The uncertainties can be quite large, depending on how one choses the prior uncertainties and how many observations there were to constrain the optimised emissions. China is a rather large region with few observation sites, similar to Brazil, Indonesia and DR Congo (those countries where CTE shows really high uncertainties in Figure 5 and 6) but if one compares the prior uncertainties with the posterior uncertainties, the optimized uncertainties have decreased. We would indeed get more information by comparing the posterior uncertainty to the prior uncertainty. However, it still reflects the

knowledge and assumptions made for the domain under study, which is an advantage of using the Ensemble Kalman Filter. Here, we show a map of observation sites constraining the inversions (we can also add this explanation to the SI, including the figure).

*4. Figure 7: It is very difficult to understand the bar of "BU Natural" and "TD Natural"? Why were the same BU anthropogenic emissions subtracted from the total emissions? Please enhance the clarity of this figure for better readability.*

The BU natural and TD natural represent the respective natural emissions from BU and TD products. As shown in Section 3.5, Table 4, not all TD products include/report the same partitions for the natural emissions. Following Reviewer's #1 suggestion and for harmonization purpose, we added only BU estimates to those TD datasets which do not include this information in their priors, and vice versa, to avoid double counting. These emissions are represented as hashed bars on top of the BU natural and TD natural columns. These hashed parts represent the missing natural partitions (Table 4).
We understand why the reviewer thinks we subtracted emissions (for example BU natural column in Russia). We only show the area which was added, and this is negative because it includes the soil sink which, according to the sign convention used in this study (negative = sink, positive = source), is represented by a negative sign.

*5. It would be great to have a figure that highlights the regions and emission sectors where the differences between current top-down and bottom-up results are most significant.*

Thank you for this suggestion. The problem is that we do not have a really complete dataset and uncertainties to be able to make such a figure. There are a lot of gaps in the UNFCCC reporting for the non-Annex I countries. We believe that Figure 7 shows rather good summary and agreement between data estimates.

*6. There are too many abbreviations in the text. It is recommended to include a table that summarizes the corresponding full forms of these abbreviations, maybe in the Appendix.*

Thank you for your observation. We made sure to explain all these abbreviations in the text, where they are introduced first. We added an explanatory table as well in the Supplementary Information file.

*7. Line 199 : missing a "."*
 Thank you, add it, now on L235

*Reference*
*Jacob, D.J., D.J. Varon, D.H. Cusworth, P.E. Dennison, C. Frankenberg, R. Gautam, L. Guanter, J. Kelley, J. McKeever, L.E. Ott, B. Poulter, Z. Qu, A.K. Thorpe, J.R. Worden, and R.M. Duren, Quantifying methane emissions from the global scale down to point sources using satellite observations of atmospheric methane, Atmos. Chem. Phys., 22, 9617–9646, https://doi.org/10.5194/acp-22-9617-2022, 2022.*

Thank you, we added the reference accordingly and cited this paper in the introduction.

---

## Author Response (AR2)

**Suggestions for revision or reasons for rejection**

(visible to the public if the article is accepted and published)

The manuscript has been improved. However, there still appear some errors or inconsistencies . Below lists some potential errors that I noted. I would suggest the authors to carefully go through the manuscript to check for plotting errors, typos, etc.

China's 2014 methane emissions (2nd UNFCCC BUR) are inconsistent between Fig. 1 (~60 Tg), Fig. 3 (~42 Tg), and L364-365 (32 Tg).

We thank the reviewer for noticing these small inconsistencies.
We checked again the numbers for China and In Figure 1 total emissions should be for the year 2014 55.29 Tg CH4 as reported by Table 2-10 page 19 second BUR. Indeed we found a unit typo for the IPPU emissions, we had 6 Tg but the correct unit was 6 kt, and we change accordingly.
We also corrected the value accordingly in Figure 3 where we show the anthropogenic excl LULUCF, and they became 53.57 Tg CH4.

We also corrected the previous years, by adapting the GWP conversion factor for CH4 from 25 (which we previously used for all countries) to 21 as reported by China.
However, in the 2$^{nd}$ Chinese BUR we note that by converting the sectoral totals from CO2eq to CH4 (GWP = 21) and summing up the sectoral values from Tables 2-13, 2-14, 2-15 and 2-16, we obtain a slightly different value than reported under line Total w/ LULUCF. We therefore might still see in the figures small differences.
We adapted the numbers for China also for all figures.

On line 374 we corrected the value to 55 Tg CH4.

The model names are inconsistent between legends of Fig. 4 and Table 2

Indeed, we refer to CAMS and MIROC4-ACTM as both runs instead of listing their full names. We changed accordingly, as the reviewer suggested, to match the model names in all figure captions and Table 2.

USA panel of Fig. 3: The mean value of BUR (black dot) is lower than the all yearly values, which is clearly incorrect. Meanwhile, the mean of value of EDGARv7 (grey) appears to be higher than most yearly values.

Thank you for this observation, we corrected the averages.

L546: "... increased emissions after '2019', ..., picks up again after '2018' ". The same thing repeated in one sentence.

Thank you, we rephrased as following: "Trend wise, all inversions agree on a slight decrease after 2013 and show increased emissions after 2018"

Russia's total emissions are inconsistent between Fig. 4 and Fig. 5.

In Figure 4 we show the time series and mean value of 2009-2021, due to the fact that TD estimates are only available after 2009. In Figure 5 we show for all the data products the average between 2015-last available year. The mean values are 12.85 Tg in Figure 4 and 13 Tg for Figure 5.

L765-769. The logic here is not very clear. More in-situ data do not provide information on "how dependent the posterior estimate is on the prior".

We agree that the in-situ data won't improve directly the posterior, however would make the prior more strong, and therefore the posterior will be better constrained by the inversion. We propose to delete the following sentence: " *Whereas the comparison of an inversion with NGHGIs or other inversions would be made more robust by having more information on how dependent the posterior estimate is on the prior.*"

As kindly agreed by the editor, we added a BU dataset, TNO_PED18-21, developed under the CoCO2 project, as input for CoCO2 WP3 (Development of global modelling and data assimilation capacity in an MVS), CoCO2 WP4 (Local and regional modelling and data assimilation) and the CO2MVS in general. It is a UNFCCC based product, for the EU27, Norway, Switzerland, UK and for Africa on the DACCIWAv2 inventory developed under CoCO2 and CAMS-GLOB-ANTv5 for other global countries. The objective was to compile a regional and global prior emission dataset for 2018 and 2021 consisting of individual components that modellers can use and that covers all relevant species and sectors, including anthropogenic, biosphere and ocean fluxes. The dataset is based on a consistent bottom-up approach at regional or global scale to ensure consistency and transparency. The TNO_CoCO2_PED appears in:
Table 1
L275, L284-288
L418-420
Figure 3 and its caption

L436-439
SI